

# Community-scale urban flood monitoring through fusion of time-lapse imagery, terrestrial lidar, and remote sensing data

Jedidiah E. Dale[1], Sophie Dorosin[1], José A. Constantine[2], Claire C. Masteller[1]

[1]Department of Earth, Environmental, and Planetary Sciences, Washington University in St. Louis, Saint Louis, MO, USA

[2]Department of Geosciences, Williams College, Williamstown, Massachusetts 01267, USA

*Correspondence to*: Jedidiah E. Dale (jed@wustl.edu)

**Abstract.** High-frequency flood events in urban areas pose significant cumulative hazards. These floods are often difficult to detect and monitor using existing infrastructure, making the development of alternative approaches critical. This study presents the implementation of a computer vision-based urban flood monitoring network deployed in Cahokia Heights, Illinois, USA.

Flood observations were collected at 30-minute intervals using consumer-grade trail cameras. Water surface elevations were estimated from the intersection of segmented flood masks with 2D-projected terrestrial lidar data. Flood extents and depths were extrapolated using a terrain depression-filling algorithm. Camera-derived peak flood extents and depths were compared to independent predictions from a 2D HEC-RAS Rain-on-Grid flood model. This procedure was applied to two flood events, one moderate and one severe, using imagery from two camera sites. For the severe event, water level estimates agreed closely

between cameras, with a median difference of less than 3 cm and a peak difference of less than 2 cm. For the moderate event, differences were larger (median <10 cm, peak <16 cm). Agreement between modeled and camera-derived peak flood extents exceeded 90% for the severe event but ranged between 21% and 42% for the moderate event. We use the convergence and divergence of independent camera observations to infer differences in spatiotemporal flood connectivity, disconnected in the moderate event and connected in the severe one. This study demonstrates the utility of low-cost, camera-based systems for

high-resolution monitoring of flood dynamics in complex urban environments and highlights their potential integration with hydrodynamic modeling.

## 1 Introduction

Flooding is the single most economically destructive natural hazard within the United States. Between 1960 and 2016, an estimated 73% ($107.8 billion USD) of direct flood property damage in the United States occurred in urban areas (National

Academy of Sciences Engineering and Medicine, 2019). The risk and impacts of urban flooding are projected to increase in coming decades, driven by climate change, expanding urban populations, and land-use change (O'Donnell and Thorne, 2019). For many regions, climate models project an increased frequency of short duration, high-intensity rainfall events, increasing flood risk in urban areas (Fowler et al., 2021). At the same time, the rate of urbanization in high flood-risk areas has outpaced other areas since 1985, increasing flood exposure risk of the general population (Rentschler et al., 2023). Together, climate





and population changes are projected to lead to an increase of 300 million people exposed to a 1% annual risk of flooding (Rogers et al., 2025).

     Both current assessments and future projections of urban flood risk frequently find significant socioeconomic disparities related to flood risk exposure both at national and local scales (Fan et al., 2025). At the national level, lower income nations are experiencing more rapid floodplain urbanization (Mazzolini et al., 2020). For individual cites, vulnerable

communities – including low-income communities and communities of color – are both exposed to more frequent flooding and experience disparate impacts (Ma et al., 2024; Selsor et al., 2023; Qiang, 2013). Neighborhood-scale differences in flood exposure, which can be driven by local differences in impervious area, microtopography, and stormwater infrastructure, are often not resolvable in metropolitan or regional-scale flood assessments (Helmrich et al. 2021; Schubert et al, 2024). Mitigating these small-scale spatial differences in flood hazards requires equivalently high-resolution monitoring of flood frequency and

intensity. Most often, this type of localized risk assessment cannot be accomplished without substantive cooperation and collaboration with impacted communities (Azizi et al., 2022).

     Pluvial flooding, which occurs when precipitation intensity exceeds local drainage capacity, can significantly impact urban environments, perhaps making it surprising that it has received comparatively less attention from researchers and policymakers (Rosenzweig et al., 2018; Prokić et al., 2019). Unlike fluvial flooding, which is typically linked to overflowing

rivers and streams, pluvial flooding is driven by local surface water accumulation, particularly during short-duration, high-intensity rainfall events (Rosenzweig et al., 2018; Azizi et al., 2022). Pluvial flooding is particularly relevant in urban landscapes, where low-lying topography and high impervious surface coverage promote rapid runoff generation (Agonafir et al., 2023). In its early stages, pluvial flooding is often characterized by spatially isolated patches of water collecting in local topographic depressions (Rosenzweig et al., 2018; Mediero et al., 2022; Cea et al., 2025). As rainfall continues, these patches

may overflow and merge, creating dynamic and expanding flood networks (Samela et al., 2020). Urban topography and infrastructure, such as roads, buildings, and stormwater systems, exert strong control on these patterns, simultaneously directing, constraining, or amplifying surface flow (Balaian et al., 2024; Beteille et al., 2025; Fan et al., 2020). Engineered drainage, such as stormwater systems, can far exceed soil infiltration in urbanized watersheds (Agonafir et al., 2023). Depending on their capacity, and condition, stormwater infrastructure can both alleviate flooding when functional but also

contribute to surface runoff when drainage capacity is exceeded (Tran et al., 2024).

     Despite its frequency and growing relevance, pluvial flooding is often excluded from traditional flood risk assessments (Rosenzweig et al., 2018; Prokić et al., 2019). Whereas fluvial flooding tends to drive large, low-recurrence events, pluvial flooding is associated with higher-frequency, lower-magnitude events – often termed "nuisance floods" because they do not typically pose an immediate threat to public safety (Rosenzweig et al., 2018). However, their cumulative socio-

economic impact over time can rival that of rare, extreme flood events, especially when the broader impacts of flood damage include transportation disruption, public health risks, and wastewater ingress into buildings (Moftakhari et al., 2017; Ten Veldhuis, 2011; Ten Veldhuis et al., 2010). In the Netherlands, the 10-year cumulative impact of smaller pluvial floods was estimated to nearly equal the damage of a single 125-year recurrence flood event (Ten Veldhuis, 2011). In the United States,



damage from pluvial floods is typically excluded from the National Flood Insurance Program, making it difficult to estimate their total economic impact (Azizi et al., 2022; National Academy of Sciences Engineering and Medicine, 2019). However, a recent study found that 87% of flood insurance claims for properties outside the FEMA-defined 100-year floodplain between 1978 and 2021 were likely related to pluvial flooding, with over 68% linked to events with less than a one-year recurrence interval (Nelson-Mercer et al., 2025). Similar findings in the United Kingdom found that 83% of reported flood damages occurred outside of designated floodplains or coastal areas, indicative of local pluvial flooding. Further, these reported damages were likely to affect properties repeatedly, highlighting the cumulative impacts of high frequency events (Dawson et al., 2008). Although data gaps remain, these recent studies provide strong evidence that pluvial flooding poses a widespread and frequently underestimated risk, motivating more comprehensive monitoring and inclusion of pluvial floods in flood risk assessments (Rosenzweig et al., 2018; National Academy of Sciences Engineering and Medicine, 2019).

Monitoring and predicting pluvial nuisance floods present distinct challenges. Traditional fluvial monitoring infrastructure, such as stream gages and water surface sensors, is not suited to detect disparate flood patches disconnected from the monitoring river system or stream (Song et al., 2024; Griebaum et al., 2017). Flood extents extrapolated from water surface levels recorded by these sensors tend to underestimate pluvially-driven flood extents, which can occur even when stream levels are below flood stage (Cea et al., 2025). To overcome these limitations, researchers have increasingly turned to distributed sensor networks to better capture the spatial heterogeneity of urban flooding (Lo et al., 2015; Song et al., 2024; Zhong et al., 2024; Mydlarz et al., 2024; Mousa et al., 2016; Azizi et al., 2022). However, both contact sensors (e.g., pressure transducers) and non-contact sensors (e.g., radar, ultrasonic) face operational challenges in urban settings, including limited installation locations and sensitivity to local disturbances (Song et al., 2024). Further, water level sensors of this nature, even when spatially distributed, only record point measurements of water level, which require further interpolation to create spatially extensive flood maps. Satellite-based and UAV remote sensing offer broader spatial coverage and, thus, have become widely used tools for flood extent mapping across a range of environments and scales (Allen and Pavelsky 2018; Tellman et al., 2021; Chanda and Hossain, 2024). However, these methods are constrained both by coarse (>1 meter) spatial and temporal (>1 day return time) resolution, making them less effective for short-duration floods and small-scale urban nuisance flood events (Tarpanelli et al., 2022; Tulbure et al., 2022; Chanda and Hossain, 2024; Zhu et al., 2022; Composto et al., 2025).

In contrast, ground-based cameras offer a promising and scalable alternative, addressing the challenges of both in-situ sensors and remote-sensing approaches (Lo et al., 2015). Ground-collected imagery provides spatially coherent measurements of flood extent within the camera's field of view, capturing continuous water surfaces in each frame, rather than isolated point readings. When deployed using consumer-grade equipment or existing infrastructure, such as traffic or security systems, cameras provide a low-cost way to achieve broad spatial coverage across urban areas (Wang et al., 2024; Lo et al., 2015). Cameras deliver high temporal resolution imagery through frequent image capture, enabling detailed tracking of flood dynamics over time. This near-continuous visual monitoring facilitates rapid flood detection and analysis, especially when combined with automated image processing techniques. Ground-collected images also capture rich contextual information,



including visible landmarks, infrastructure, and human activity, enhancing the interpretation of flood impacts and supporting more comprehensive urban flood management.

There are three broad approaches to camera-based flood monitoring. The first and most developed relies on identifying water levels relative to known benchmarks, such as topographic markers or staff gauges. This approach is particularly well-suited to river or reservoir environments, where stage progression and flooding is more predictable (Sabbatini et al., 2021; Chapman et al., 2022, 2024; Johnson et al., 2025). However, its applicability can be limited when flood extents are irregular or spread over complex urban terrain without extensive available benchmarks from which water levels can be derived. A second approach uses the fraction of an image classified as flooded to estimate water level and extent. This method
requires the development of a quantitative correlation between flooded image fraction and water level (de Vitry et al., 2019; Vandaele et al., 2021). Hybrid methods combine both techniques, using image segmentation and reference objects to estimate flood depths. For example, Vandaele et al. (2021) used surveyed landmarks to constrain absolute water level. Liang et al. (2023) used the automated identification of street signs and humans in flood images to estimate water depth. While shown to be promising, these methods often depend on stable camera positions, consistent lighting, and persistent ground control points.

A key limitation of image-only flood monitoring approaches is their difficulty in translating two-dimensional image pixel data into real-world flood depths, particularly in heterogeneous urban landscapes where water may accumulate in shallow, discontinuous patches. More advanced methods have addressed this challenge by integrating camera imagery with high-resolution topographic data, such as lidar or Structure-from-Motion (SfM) (Wang et al., 2024; Griesbaum et al., 2017). Pairing high-resolution topographic data products with known camera geometries allows floodwater-identified pixels to be
geo-referenced and intersected with the underlying terrain, yielding spatially distributed flood depths even in settings where flood boundaries are irregular or flood waters evolve rapidly (Erfani et al., 2023; Eltner et al., 2018, 2021). As a result, these methods can overcome the spatial ambiguities inherent in image-based approaches and are particularly valuable in the complex topography of urban environments. However, most demonstrations of this technique have occurred in controlled or short-term deployments with stable cameras and ground control, with the notable exception of Blanch et al. (2025), which successfully
applied projection-based stream level estimation over a continuous two-year period. The feasibility of its use for long-term flood monitoring in urban environments, with limited ground control, and frequently changing scene context, requires additional research.

Camera-based flood monitoring approaches ultimately rely on robust image segmentation to accurately identify water presence and extent. Traditional image processing techniques such as intensity thresholding and random-forest classification
remain useful in structured environments (Chapman et al., 2024; Lo et al., 2015; Griesbaum et al., 2017), but most recent work has transitioned towards deep learning-based semantic segmentation models to classify water pixels (Erfani et al., 2023; Eltner et al., 2021; Wang et al., 2024). Architectures range from U-Net models (Vitry et al., 2019), to more complex vision transformers (Erfani et al., 2023; Zamboni et al., 2025), with many models achieving high segmentation accuracy when trained on domain-specific datasets. Indeed, a systematic evaluation of 32 network architectures trained on the same dataset of river
images demonstrated the efficacy of the method, with 24 of the tested architectures achieving greater than 90% testing accuracy



(Wagner et al., 2023). The recent emergence of foundation models such as Segment Anything (SAM) (Kirillov et al., 2023; Ravi et al., 2024) has introduced the possibility of domain-agnostic water segmentation. Recent studies have demonstrated that with minimal fine tuning, these domain-agnostic foundation models can achieve comparable classification accuracy to state-of-the-art, domain-specific models (Moghimi et al., 2024; Wang et al., 2024).

In this study, we present a flexible and operationally oriented framework for monitoring urban flood extent and depth using time-lapse imagery from ground-based cameras in combination with terrestrial and aerial lidar data. Our implementation focuses on a community in Cahokia Heights, IL experiencing chronic pluvial flooding within the Mississippi River floodplain. Using consumer grade trail-cameras, we demonstrate accurate, centimeter-scale, water level estimation, and flood extent extrapolation, for two case study flood events. Our approach emphasizes adaptability to site conditions, minimal reliance on

fixed benchmarks, and limited need for long-term infrastructure. As an independent benchmark, we compare our camera-derived estimates with output from a two dimensional, HEC-RAS rain-on-grid hydrodynamic model, evaluating the efficacy of camera-based monitoring in operational flood modeling workflows.

## 2 Methods

### 2.1 Study site

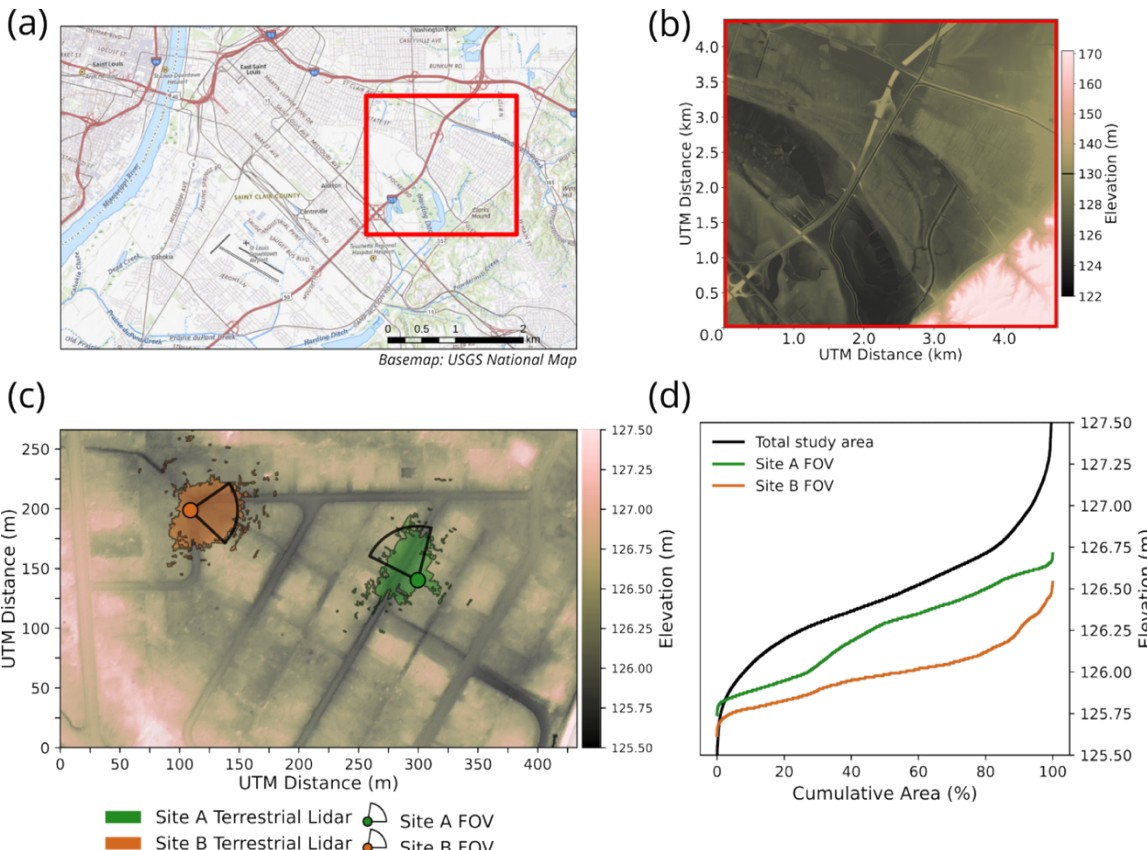




**Figure 1: (a) Study location of Cahokia Heights, IL (b) broader study area shaded relief (c) Terrestrial lidar point cloud extent and camera monitoring location field of view (FOV) (d) Cumulative elevation distributions (hypsometric curve) for the bare-earth lidar elevations for entire study area (black), and camera FOVs (orange and green, respectively).**

Flood monitoring efforts were conducted in collaboration with the Cahokia Heights community, located in St. Clair
County, Illinois, within the Mississippi River Floodplain (Figure 1a). Cahokia Heights residents have long experienced chronic nuisance flooding, driven primarily by pluvial processes (U.S. EPA, 2021; Maganti, 2020; Colten, 1988; Schicht, 1965). Despite the occurrence of multiple impactful floods each year, the entirety of area presented in this study falls outside of FEMA defined Special Flood Hazard Areas (FEMA, 2003). This frequent flooding is attributable to both natural and engineered factors. The region's clay-rich floodplain soils exhibit poor drainage, and in combination with the low-relief landscape (Figure
1b-d), water readily accumulates in surface depressions (USACE, 2023). Compounding these natural vulnerabilities, decades of infrastructure neglect have left the sewer and stormwater systems in disrepair. Many of the existing sewer and stormwater pipes are undersized or blocked, reducing drainage capacity and leading to recurrent sewage backups and drinking water contamination (USACE 2024). As a result, persistent flooding has caused significant property damage, disrupted the daily activities of residents, and compromised household plumbing systems (Musiker et al., 2021). At the time of this study, no
formal flood monitoring infrastructure existed within the community.

## 2.2 Community-scale monitoring network

To begin addressing this gap, eight time-lapse camera flood monitoring stations were installed in the fall of 2020 in collaboration with community residents. This study focuses on two of those stations, herein Sites A and B (Figure 1b). Cameras A and B are located on opposite sides of a residential neighborhood, approximately 190 meters apart, with non-overlapping
fields of view. Camera A is positioned on a straight stretch of road on the eastern side of the neighborhood at an elevation of 126.1 m (NAVD88). Camera B is located on the western side of the neighborhood, at a slightly lower elevation of 125.95 m, placed at a slight bend in the road (Figure 1c). Each monitoring station consists of a Blaze A52 trail-camera (16 MP, $f$4 mm) in a transparent-faced plastic housing mounted approximately 1.5 m off the ground on metal conduit pipe driven into the soil. Each camera is set to capture a 5,120 by 2,880-pixel resolution image and a five second video every 30 minutes. Due to
excessive glare during nighttime operations, the cameras' infrared flash was disabled, and ambient lighting from streetlights was used for nighttime operation. Images were retrieved approximately every two months during site visits, during which batteries and SD cards were replaced. Camera disturbances, motion, or damage were documented at each service interval.

This study draws on two primary sources of topographic data: a regional aerial lidar survey and terrestrial lidar acquired at each camera site in 2023. The aerial lidar dataset was collected across St. Clair County in 2019 as part of the USGS
3D Elevation Program (3DEP) and the Illinois Height Modernization Program (ILHMP) (Aerial Services Inc, 2021; USGS, 2022;). LAS-format point clouds were obtained for the study area, with an average point density of 4.0 points per m². These data were interpolated into a 0.5-meter resolution digital terrain model (DTM) using inverse distance weighting (IDW), implemented via the Point Data Abstraction Library (PDAL) (PDAL Contributors, 2022). Additionally, a void-filled DTM at



the same resolution was created using triangulated irregular network (TIN) interpolation. This processed product is referred to
throughout the paper as the USGS DTM.

While aerial lidar offers broad spatial coverage, it does not resolve fine-scale topographic features such as street curbs or shallow depressions common in urban environments. To capture these features, the aerial dataset was supplemented with high-density terrestrial lidar data collected in June 2023 using a Zeb Horizon GeoSLAM handheld scanner. The scanner emits 300,000 near-infrared (930 nm) laser pulses per second, with an operational range of up to 100 meters and a reported point
accuracy of approximately 6 mm (FARO, 2024). At each camera site, terrestrial scans covered an area of approximately 2,000 m² and achieved point densities exceeding 1,000 points per m², with upwards of 100,000 points per m² in the center of the survey area. Ground points were classified using a cloth simulation filter (CSF) in PDAL (Zhang et al., 2016), and bare-earth DTMs were interpolated at a 0.5 m resolution.

To enable geospatial integration with the USGS DTM, five to six reflective ground control points (GCPs) were
deployed at sites A & B during each scan and surveyed with an Emlid Reach GNSS receiver. The location of each camera post was also surveyed and used as a GCP. Paired GCP survey locations ($E_{UTM}$, $N_{UTM}$, $Z_{NAVD88}$) and the corresponding raw point-cloud coordinates ($X_{GCP}$, $Y_{GCP}$, $Z_{GCP}$) were used to compute a rigid transformation matrix ($\mathbf{P_{GCP}}$) for each scan. Given the limited extent of each site, rotations about the $x$- and $y$-axes were neglected. Each terrestrial point cloud was then projected into a common coordinate system (NAD83/UTM Zone 15N) using the NAVD88 vertical datum. This transformation aligned
the terrestrial lidar data with the USGS DTM, enabling direct elevation comparisons between the regional and site-specific models. The accuracy of co-registration was assessed by calculating vertical differences between the two DTMs at 0.5-meter resolution. Corrected GCP locations had a lateral Root Mean Square Error (RMSE) of 2.0 cm at Site A and 7.5 cm at Site B. Median elevation differences relative to the USGS DTM were 1.5 cm and 3.7 cm near road surfaces at Sites A and B, respectively.

**2.3 Case study flood events**

The monitoring methodology was applied to two flood events in 2024: a moderate severity event on 14 May 2024, and a high-severity event on 04 July 2024. All times are given 24-hour Central Daylight Time (UTC-5). Image capture times for Cameras A and B were offset 14 minutes for the May event, and 9 minutes for the July event. The moderate severity flood event was triggered by 12 mm of rainfall over an 8-hour period. At Camera A, flooding was documented in 9 total images,
capturing two distinct flood pulses. The first pulse was approximately two hours in duration, appearing in 4 consecutive images, followed by a 1.5-hour dry interval spanning three images, and then a second pulse observed across five images, for a 2.5-hour duration. During the early phase, the water surface remained below the ~20 cm street curb, resulting in partial shadowing and limited visibility of the flood extent. Peak inundation at Camera A was observed during the second pulse, at 17:51, when floodwaters overtopped the curb and extended into residential yards. At Camera B, the May 14 flood was captured in 13
consecutive images, beginning at 14:05 and persisting through 20:05. Floodwaters appeared to peak at approximately 14:35,



completely inundating the roadway and advancing into adjacent yards. The continuous visibility of standing water throughout the observation period suggests sustained surface accumulation, characteristic of ineffective drainage during moderate rainfall.

The more severe, 04 July 2024 event that followed occurred in response to 82 mm of rainfall over 11 hours, preceded by an additional 10 mm of antecedent rainfall on 03 July 2024. Based on NOAA duration-frequency curves, this precipitation

event corresponds to approximately a four-year recurrence interval (NOAA OWP, 2025). At Camera A, flooding was observed in 20 consecutive images, spanning from 08:45 to 17:45. Peak inundation occurred at 12:45, when floodwaters reached residential porches and exceeded the camera's field of view. At Camera B, visible flooding began at 04:54 and was documented in 32 images, ending at 20:24. At its observed peak at 12:54, floodwaters completely flooded yards and reached the foundations of multiple homes.

**2.4 Water segmentation from images**

(a) Flood event image    (b) 2D image reference points    (c) 3D lidar reference points

Water Pixel Segmentation

Camera Pose Estimation

(d) Binary flood mask    (e) Image projected point cloud

Georeferencing Transform

(f) Visible flood extent    (g) Water surface elevation    (h) Total flood extent

Canny Edge Detection

Flood Fill Propagation



**Figure 2: Procedure for estimating flood extent from a flooded image at Site B. Image and point cloud reference features are used to estimate camera pose and project points onto the image plane. The intersection with the flood mask gives the visible flood extent. Water surface elevation (WSE) is extracted from the edge elevations and propagated for the final, total flood extent.**

Flooded pixels in each time-lapse image were segmented using SegmentAnything2 (SAM2) (Kirillov et al., 2023; Ravi et al., 2024) (Figure 2). Classifications were made using a pre-trained set of model weights. Image sequences were processed as videos to facilitate the tracking of identified flood regions across successive frames. Because SAM2 is not explicitly trained for water segmentation, a manual prompting approach was used, similar to the SAM-Six-Point method described by Zamboni et al. (2025). This approach relies on annotated point prompts that indicate the presence or absence of

flooding at individual pixels within a reference image.

      For a given flood event, the earliest image in which flooding was visible was annotated with three to five positive point prompts. These prompts were then used to segment the remaining image sequence. The visual confirmation of flooding was used to iteratively refine the segmentation, with additional positive prompts added to correct for false negatives (i.e., flooded areas classified as non-flooded), and negative prompts added to address false positives (i.e., non-flooded areas

misclassified as flooded). This process continued until flood extents were satisfactorily delineated based on visual agreement with apparent surface water boundaries. The final output of this classification procedure is a binary flood mask for each image, where pixel values of one indicate flooded regions and values of zero indicate non-flooded areas. SAM2-predicted flood masks were evaluated against manually labeled flood extents to quantify segmentation accuracy using the Intersection-over-Union (IoU) metric. IoU is defined as the ratio of true positive water-classified pixels to the total of all true positives, false positives,

and false negative pixels.

      In addition to spatial classification, each flood mask was used to calculate a relative measure of flood severity per image. This was quantified as the flooded pixel fraction, or the number of pixels classified as water divided by the total number of pixels in the image. This ratio is referred to as the Static Observer Flooding Index (*SOFI*), following the approach of Vitry et al. (2019), providing a simple proxy for flood intensity as seen from a fixed observation point.

**2.5 Point cloud to image projection**

      The workflow for estimating floodwater elevation and extent relies on establishing a correspondence between features visible in time-lapse images and their three-dimensional coordinates within a georeferenced terrestrial point cloud. This correspondence requires knowledge of both the intrinsic parameters of each camera (such as focal length and sensor dimensions) and its extrinsic parameters, which describe the camera's location and orientation in space, referred to as the

camera pose. The projection of a three-dimensional point cloud ($X$, $Y$, $Z$) in world coordinates onto a two-dimensional pixel coordinate on the image plane ($u$, $v$), is defined by Equation (1):

$$\begin{bmatrix} u \\ v \\ 1 \end{bmatrix} = \mathbf{KP} \begin{bmatrix} X \\ Y \\ Z \\ 1 \end{bmatrix}, \qquad\qquad (1)$$



where

$$\mathbf{P} = [\, \mathbf{R} \mid t \,]. \tag{2}$$


      This projection is governed by two key transformation matrices: the intrinsic matrix, **K**, a 3×3 matrix that encodes the camera's internal geometry, including focal length and optical principal point, and the extrinsic matrix, **P,** which combines a rotation matrix, **R**, and translation vector, **t**, to describe the camera's pose relative to the world coordinate frame, as shown in Equation (2). Together, these matrices enable transformation from world coordinates into image space.

Intrinsic parameters for each camera were estimated in a controlled, laboratory-based calibration using a checkerboard target with 25 mm squares printed on 216 mm by 279 mm paper. Between 20 and 30 images of the target were collected from multiple oblique angles. Collected images were processed using OpenCV, a standard computer vision library (Bradski, 2000), identifying checkerboard corners, computing the intrinsic matrix, **K**, and estimating a five-element distortion coefficient vector, **d**. This distortion matrix is used to correct projected pixel coordinates to improve accuracy of point to image projection.

The extrinsic camera pose matrix, **P**, was estimated based on a set of matched reference features with known locations in both image coordinates $(u, v)$, and world coordinates $(X, Y, Z)$. This process, known as the Perspective-n-Point (PnP) problem, yields an estimated camera pose denoted as $\mathbf{P_{PnP}}$. Feature matching was performed manually, with image coordinates of reference features labeled in ImageJ (Schindelin et al., 2012) and their corresponding world coordinates annotated from the terrestrial lidar point cloud using CloudCompare (CloudCompare, 2023). In the absence of permanent ground control points,

static scene elements such as rooftops, fence posts, and utility poles were used as reference features. Between 20 and 30 such features were labeled for each camera. Point precision was limited by image resolution, point cloud noise, and the spatial resolution of the lidar scan.

      Using these reference features, we estimated $\mathbf{P_{PnP}}$ using the Efficient Perspective-n-Point Camera Pose Estimation (EPnP) algorithm (Lepitit et al., 2009) as implemented in OpenCV. A random sample consensus (RANSAC) procedure was

applied iteratively solve for the optimal extrinsic camera post matrix, $\mathbf{P_{PnP}}$. For each of 10,000 random sub-samples of labeled reference points, $\mathbf{P_{PnP}}$ was computed and evaluated by its reprojection error, defined as the Euclidean distance between each labeled image coordinate $(u_r, v_r)$, and the associated projected world coordinate of the feature $(u_{rp}, v_{rp})$. Points with a reprojection error exceeding 50 pixels were classified as outliers. The RANSAC iteration minimizing the number of outlier points was selected as the optimal camera pose matrix estimate, $\mathbf{P_{PnP}}$. Accuracy of the estimated camera locations was validated

by comparing the recovered camera location with the known camera position extracted from the terrestrial lidar dataset. Additional laboratory experiments were conducted to verify the performance of this workflow (Supplementary Information).

      The estimated camera pose matrix, $\mathbf{P_{PnP,}}$ is initially referenced to an arbitrary, local coordinate system of the terrestrial point cloud. To enable projection from topographic coordinates into image space, $\mathbf{P_{PnP}}$ was composed with the inverse of the rigid-body transformation matrix $\mathbf{P_{GCP}}$, obtained during georeferencing, as Equation (3):



$$\begin{bmatrix} u \\ v \\ 1 \end{bmatrix} = \mathbf{K} \mathbf{P_{PnP}} \mathbf{P_{GCP}^{-1}} \begin{bmatrix} E \\ N \\ Z \\ 1 \end{bmatrix} \qquad (3)$$

Equation (3) represents a final transformation pipeline that maps UTM/NADV88-referenced coordinates ($E$, $N$, $Z$) to corresponding image pixel coordinates ($u$, $v$). Using this framework, each point in the terrestrial lidar point-cloud is assigned a pixel location, enabling spatially coherent visualization and quantification of flood extent in the camera imagery (Figure 2).

A separate camera pose estimate was computed for each camera and flood event. For the moderate May 14 flood,
Camera A's pose was calculated using 18 reference features, yielding a median reprojection error of 6.83 pixels. The recovered camera location was offset 46 cm from the labeled camera center in the point cloud. For the July 4 event, pose estimation at Camera A used 24 features, resulting in a median reprojection error of 23.6 pixels and a reduced camera position offset to 6 cm. For Camera B, the May 14 pose was calculated using 22 reference features, resulting in a median reprojection error of 4.99 pixels. Due to camera movement following the lidar survey, estimated positional uncertainty can only be resolved as less
than 1 m. The pose estimate at Camera B for the July 4 event used 16 reference features, yielding a reprojection error of 8.9 pixels, again with a positional uncertainty below 1 m.

## 2.6 Flood extent estimation

Flood extent estimation is based on the intersection of lidar-derived topography and image-derived water classifications. Using the established projection pipeline in Equation 2, each point in the terrestrial lidar point cloud is mapped
to a corresponding image pixel. If a pixel is identified as flooded in the SAM2-derived binary segmentation mask, the associated terrestrial lidar point is classified as inundated. This set of inundated points represents the portion of the ground surface that is underwater at the time the image was captured. These inundated points are interpolated into a 0.05-meter resolution raster covering the visible flood extent in the image. To estimate water surface elevation (WSE), the highest elevations along the boundary of the inundated zone are used as a proxy for the maximum water level and the water surface is
assumed to be flat. Edge pixels are extracted using a Canny Edge Detection filter, and the 90th and 95th percentiles of the extracted edge elevation distribution are used to represent a range of possible water surface levels ($WSE_{90}$ and $WSE_{95}$) to account for potential topographic noise or obstruction of the water edge in the time lapse images.

To estimate flood extent beyond the visible portion of the image, we apply an iterative flood-fill procedure to the 0.5 m-resolution USGS DTM (Wu et al., 2018; Samela et al., 2020). Beginning at the lowest observed elevation within the camera's
field of view, adjacent terrain cells are iteratively inundated if their elevation is below the target WSE, continuing until no additional cells meet this condition. The area of interest for the flood-fill implementation focused on the direct area spanning the two camera locations, approximately 500 m by 250 m, to avoid propagation into unobservable areas. This approach assumes no-flow resistance and instant water propagation. The resulting inundated area is then converted into a flood depth map by subtracting the DTM elevation from the estimated WSE. Repeating this process for each timestamped image yields a time
series of inundation maps at 30-minute intervals and 0.5 m spatial resolution for each camera site. We perform this propagation





independently for each event at each monitoring site to assess the potential variability in estimated WSE derived from monitoring sites with distinct scene geometries and fields of view.

**2.7 Comparison to a pluvial flood model**

Image-derived flood extents and depths were benchmarked against results from a two-dimensional pluvial flood
model. This model is implemented using the Hydrologic Engineering Center's River Analysis System (HEC-RAS), configured with a "rain-on-grid" unsteady boundary condition to simulate overland water flow across an 89.6 km² model domain covering the study site (USACE, 2022). The base terrain is the 0.5 m USGS DTM. Rainfall is uniformly applied to the domain, and water movement is governed by the diffuse-wave approximation of the shallow water equations. For the 14 May 2024, flood event, precipitation inputs were sourced from the station at St. Louis Downtown Airport (NWS:KCPS), approximately 6 km
from the site. For the 04 July 2024, event, rainfall records from the USACE Mississippi River Station (USACE:ENGM7), located 8 km away, were used. Storm drain locations and connectivity assumptions are based on a survey by the Illinois Department of Natural Resources (IDNR, 2023; Heartlands Institute, 2023). Model roughness values are informed by National Land Cover Database (NLCD) (USGS, 2023) classifications and refined using road and building footprint data (Illinois State Water Survey, 2018). Model outputs were generated at the same spatial and temporal resolution as the image-derived flood
datasets to enable direct comparison. Although image data informed general model development, no direct calibration against the imagery was performed.

To compare modeled and image-derived flood extents across consistent spatial scales, analyses were conducted at two levels: the neighborhood study area and the individual fields of view (FOV) for each camera (Figure 1b). FOV for Camera A during the moderate (severe) flood are estimated to be roughly 1,646 m$^2$ (1,387 m$^2$), with terrain elevations ranging from
125.74 m (125.73 m) to 126.84 m (126.84 m). For Camera B, the field of view area is estimated as 1,442 m$^2$ (1,360 m$^2$) with an elevation range of 125.61m (125.56 m) to 126.56 m (126.53 m) for the moderate (severe) flood event. Differences in the total field of view reflect a 13.1° shift in orientation at Camera A and a 23.6° shift in orientation at Camera B between flood events. These differences in spatial coverage and viewing geometry are important for interpreting agreement or disagreement between modeled and image-derived flood extents. As such, we use distinct spatial footprints areas for each model-data
comparison.

Our comparison focuses on quantifying the relative agreement in predicted flood extent between the two methods. The primary metric focuses on identifying regions where both the model and camera-based approaches indicate flooding – areas of mutual agreement in predicted inundation. This shared extent is expressed as $F_{overlap}$, the ratio of the number of pixels classified as flooded by both methods to the total number of pixels classified as flooded by either. The model domain includes
areas separated from our camera sites by major roads and drainage canals. To provide a meaningful comparison between model output and our image-based methods, we spatially restricted our comparison to a region with the approximate bounds of the topographic depression containing the study neighborhood. Where flood extents overlap, we also compared modeled and observed water surface elevations and flood depths.



# 3 Results

## 3.1 Visual flood observations


Flood segmentation performed robustly across both monitoring sites and flood events. Following refinement using additional point prompts, image segmentation and classification produced accurate flood masks with strong agreement with manually labelled flood extents. For the moderate May 14 event, the mean intersection-over-union (IoU) for Site A was 91% (range 21%) and 93% (range 18%) for Site B. For the severe July 4 event, mean IoU was 90% (range of 14%) at Site A and

93% (range 23%) at Site B. Refinement prompts successfully eliminated all whole-image false positives. Most discrepancies occurred when flood waters were partially occluded (e.g., by vegetation, fences, or vehicles) or where strong reflections caused misclassification. Fuzziness in flood boundaries increased further away from the camera, as pixel ground resolution decreased (Eltner et al., 2021).

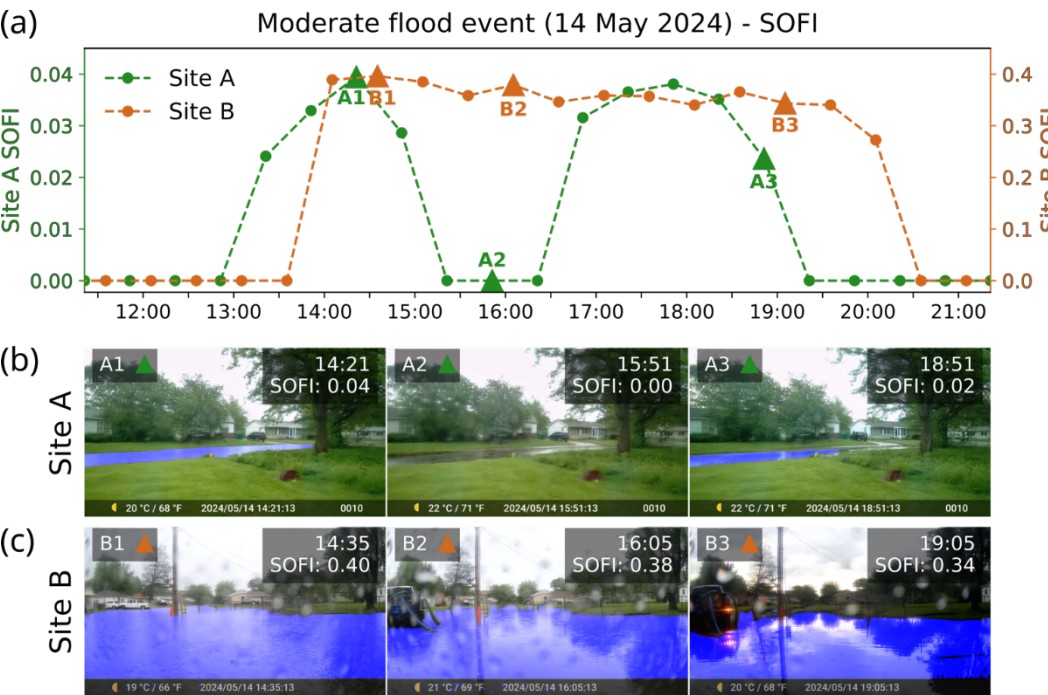

**Figure 3: (a) *SOFI* time series for 14 May moderate severity case study event. Representative flooded images from (b) Site A and (c) Site B. Segmented flood masks are shown in blue.**



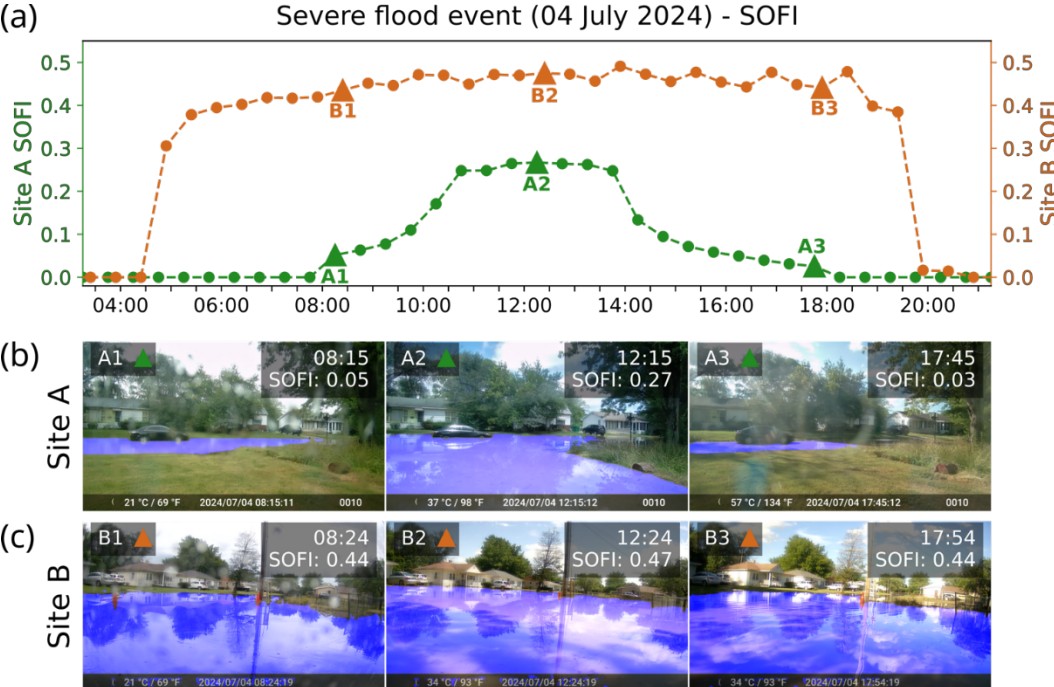

**Figure 4: (a)** *SOFI* **time series for 04 July severe case study event. Representative flooded images from (b) Site A and (c) Site B. Segmented flood masks are shown in blue.**

The segmented flood masks were used to quantify the spatial extent of visible flooding using the Static-Observer Flood Index (*SOFI*) (de Vitry et al., 2019) (Figure 3, 4). During the moderate May 14 event, two distinct flood pulses are observed at Site A, both with peaks in *SOFI*=0.04 (Figure 3a). These pulses are separated by dry conditions where *SOFI*=0. In contrast, *SOFI* values at Site B were consistently non-zero, indicating persistent flood inundation for the entire duration of the observational period. Peak *SOFI* at Site B during the moderate event is 0.40. At both sites, the range of *SOFI* during the

July 4 severe flood is elevated compared to the moderate flood event, reflecting a wider range of water surface elevations imaged. During the rising limb of the severe flood at Site A, *SOFI* increased steadily from 0.05 to 0.27, where it stays within a range of 0.002 for 1.5 hours, before declining monotonically to 0.025 as floodwaters receded (Figure 4a). Values are higher for the entirety of the event at Site B, with *SOFI*=0.31–0.49. However, compared to Camera A, changes in *SOFI* during the flood event are more muted at Site B. These differences in *SOFI* magnitude and variability are likely driven by differences in

scene geometry. At Site A, the camera is positioned farther from, and perpendicular to, the road capturing a broader view that includes a resident's lawn in the foreground (Figure 3b, 4b). In contrast, Site B's camera has a tighter field of view focused exclusively on the road surface (Figure 3c, 4c). Although flooding begins on the road in both locations, the prominence of the roadway in Site B's imagery makes the flooding more visually dominant in the scene. These results demonstrate that *SOFI* provides a reliable metric for tracking relative changes in inundation at individual sites and accurately captures the timing and

progression of pluvial flooding. However, a direct comparison of *SOFI* values between monitoring sites, and flood events, is complicated by variations in camera placement, viewing angle, and scene composition.



## 3.2 Water levels and flood extents

Using the final projection pipeline in Equation (3), with a distortion correction applied (See Supplementary Information), water surface elevation (WSE) time series were estimated at 30-minute intervals at each monitoring site for each

flood event. We report both $WSE_{90}$ and $WSE_{95}$, representing the 90th and 95th percentiles of water surface edge elevations.

During the May 14 flood event, $WSE_{90}$ peaked twice at Site A, during each of the distinct flood pulses in the *SOFI*-derived hydrograph (Figure 5a). $WSL_{90}$ rose from 126.00 m at the onset of visible flooding, to peaks of 126.06 m and 126.04 m, separated by a period of no-flooding for all images with *SOFI*=0. Following the second peak, water level declines to a minimum resolved water level of $WSE_{90}$=125.94 m. These observations yield a range of image-derived water surface elevations

of 11.3 cm. $WSE_{95}$ at the same site ranged from 126.02 m to 126.13 m, peaking on the first and last images, reflecting a comparable 11.7 cm rise. In contrast, Site B experienced a broader range of water levels of approximately 34.5 cm m, with $WSE_{90}$=125.84 to 126.18 m), and $WSE_{95}$=125.88 to 126.23 (range=35.6 cm), reflecting a more continuous rise and fall in water levels, rather than then distinct pulses observed at Camera A.  Water levels at Sites A and B differed by a mean of 9.2 cm for $WSE_{90}$ and 12.8 cm for $WSE_{95}$, supporting the interpretation that the floodwaters occupied two disconnected patches, filling

independently over the course of the event. Water level ranges for Sites A and B overlap for only a single image pair. Water level sensitivity to the elevation percentile used is similar between sites, with median ranges between $WSE_{90}$ and $WSE_{95}$ of 2.5 cm  and 3.6 cm, for Sites A and B. Ranges between $WSE_{90}$ and $WSE_{95}$  were highest at low water levels, particularly at Site A, where maximum ranges of 13.1 cm and 19.0 cm occurred in the first and last images, compared to an average of 2.4 cm for the remaining images. At Site B, these ranges were generally smaller and more consistent, ranging between 1.8 cm and 7.6

cm, with an average of 3.9 cm.



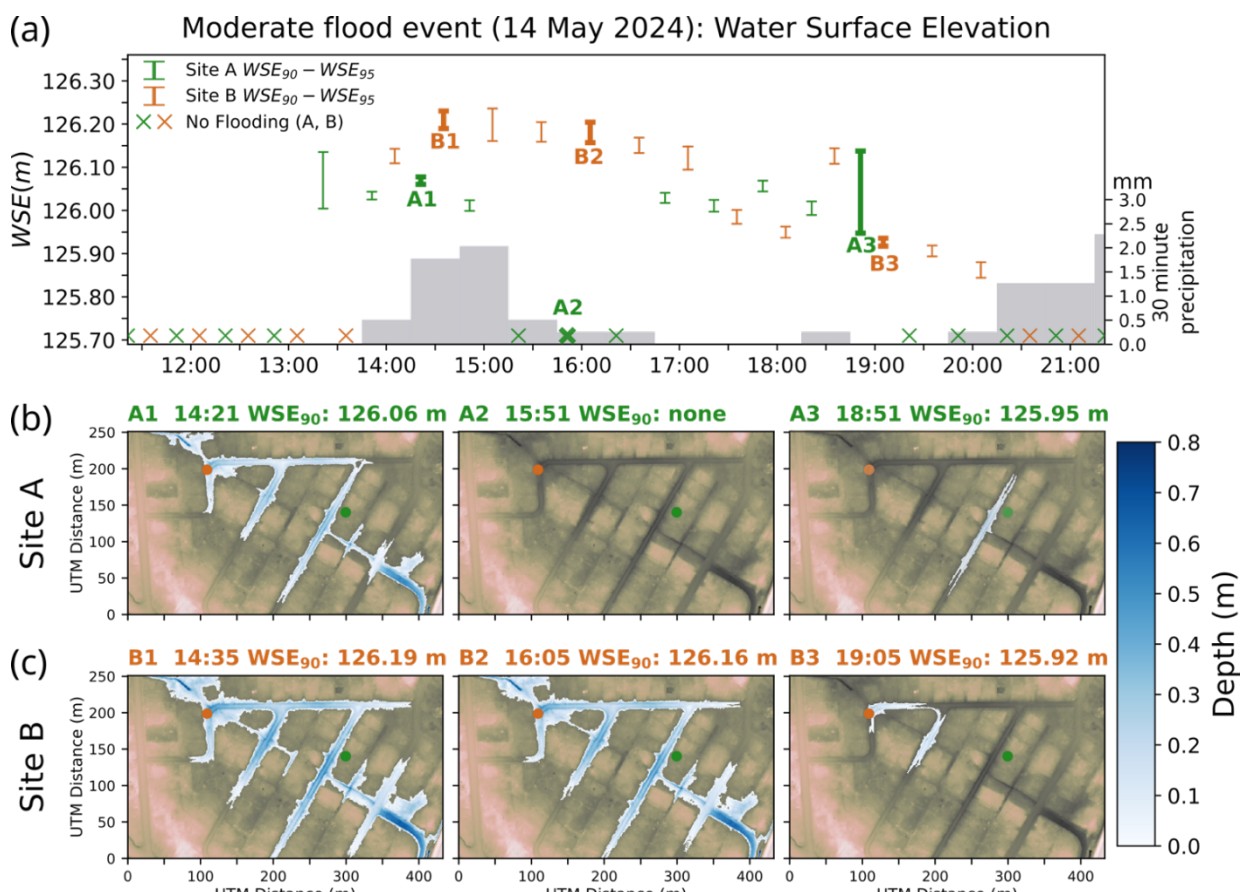

**Figure 5: (a) Water surface elevations (WSE) estimated from image-lidar projection for the 14 May case study event. Flood-fill extents propagated from $WSE_{90}$ at (b) Site A and (c) Site B.**

During the more severe July 4 flood, water surface elevations rose substantially at both sites, characterized by clearly
defined rising and falling limbs in both the *SOFI* and camera-derived hydrographs (Figure 6a). At Site A, $WSE_{90}$ varied by
over 35.7 cm and $WSE_{95}$ over 27.9 cm, with peak water levels between 126.30 m and 126.32 m. At Site B, both $WSE_{90}$ and
$WSE_{95}$ spanned a larger vertical range of 53.2 cm, with peak levels overlapping closely with those at Site A (126.28 to 126.30
m), matching the peaks in the *SOFI* hydrograph within 30-minutes. The difference in maximum water surface elevations
between sites were just 1.7 cm and 1.0 cm, strongly suggesting that floodwaters formed a single, hydraulically connected
inundation zone spanning both monitoring locations. Compared to the May event, the range between $WSE_{90}$ and $WSE_{95}$ during
the July flood was more consistent over time, with Site A showing a lower and more stable mean difference of 2.3 cm
(range=7.9 cm), versus a mean difference of 3.9 cm (range=19.2 cm) at Site B. As in the earlier flood, uncertainty was greatest
at the beginning and end of the event, when lower water levels produced a wider spread between $WSE_{90}$ and $WSE_{95}$. Water
levels between sites agreed most closely on the rising limb of the flood, with WSE ranges overlapping across nearly every
time step leading up to the peak. In contrast, the more gradual recession of floodwaters at Site B resulted in increasing



divergence during the falling limb. Despite these discrepancies, the rate and direction of change in water level remained largely consistent between cameras.

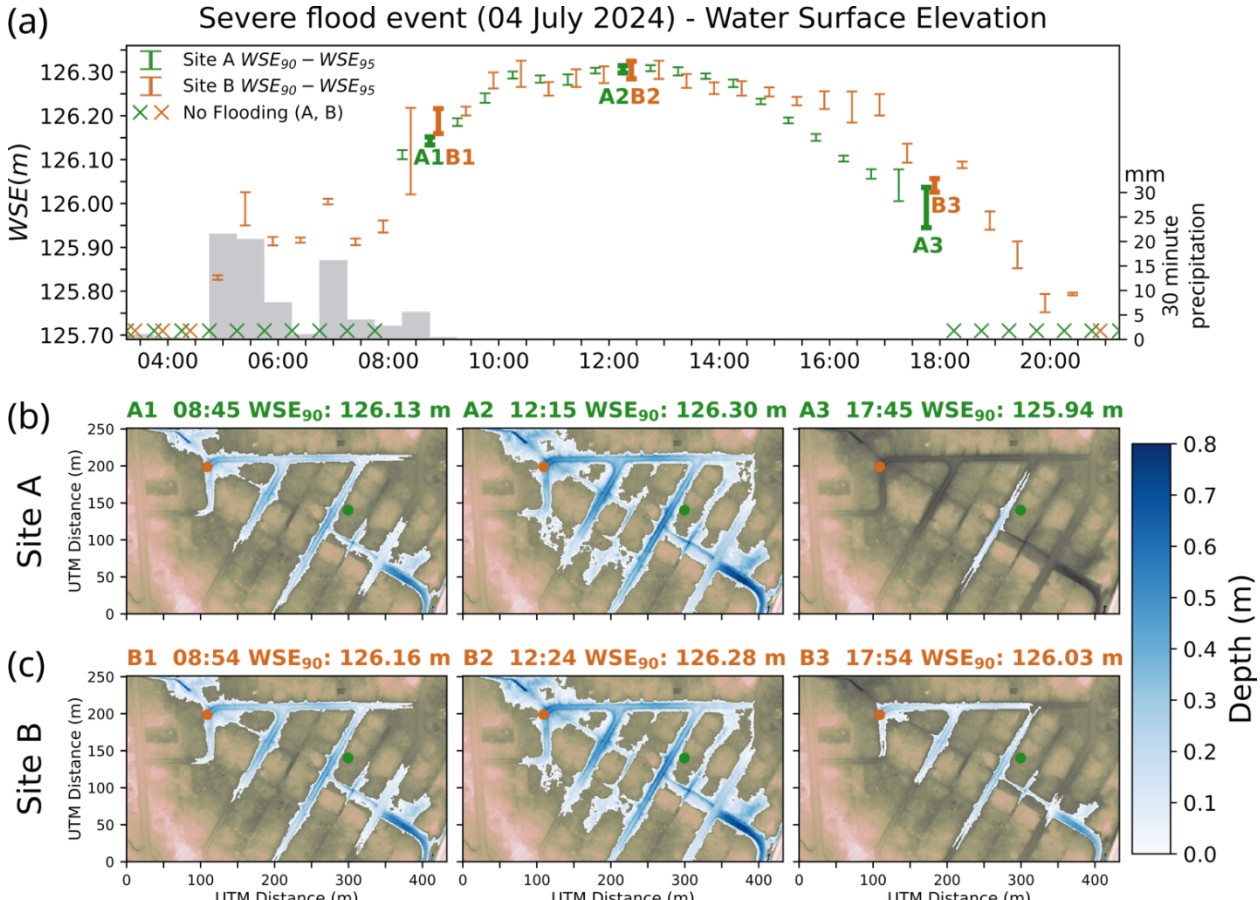

**Figure 6: (a) Water surface elevations estimated from image-lidar projection for the 04 July case study event. Flood-fill extents**
**420    propagated from $WSE_{90}$ at (b) Site A and (c) Site B.**

Differences in water surface elevation between the May and July flood events directly influenced the connectivity and extent of resulting inundation, with important implications for interpreting flood dynamics. Flood-fill propagation using $WSE_{90}$ values from Site A produced spatially restricted inundation, with a maximum inundated area of $1.1 \times 10^4$ m², with flooding largely confined to a patch near Camera A, only connecting to Site B at each peak in $WSE_{90}$ (Figure 5b,c). In contrast,
when flood-fill is propagated using the higher $WSE_{90}$ values from Site B, inundation extends to Site A until part way through the flood recession, including the period with no visible flooding at Site A (Figure 5b). The resulting maximum flood extent derived from Site B $WSE_{90}$ was $2.0 \times 10^4$ m², exceeding the Site A–based extent by 54.7%. Maximum flood depths propagated using the Site B-derived hydrograph were 9.8% greater than Site A, equivalent to 0.13 m. The maximum difference in propagated flood extents was $1.6 \times 10^4$ m² (164%), driven by an 11.9 cm difference in $WSE_{90}$ between Sites A and B. These
differences resulted in a 91 cm m variation in maximum flood depth, with flood extents based on Site B's $WSE_{90}$ values





producing inundation depths that were 102% greater. These differences are consistent with the presence of two distinct flood pulses at Site A and a more gradual and persistent rise at Site B. The significant difference in $WSE_{90}$ between the sites supports the interpretation of transient connectivity, with sensitivity to threshold water levels contributing to large differences in mapped extent.

In contrast, during the July 4 flood event, peak WSE90 values at Sites A and B differed by only 1.7 cm (Figure 6a), and flood-fill propagation from either site resulted in qualitatively similar inundation patterns (Figure 6b, c). Both flood-fills generated a single, continuous inundation zone extending across the low-lying area between sites for most of the flood event. Propagation from Site A produced a peak extent of $3.2\times10^4$ m², while propagation from Site B produced a peak extent of $3.0\times10^4$ m², a difference of 6.2%. Over the entire event, Site A and B propagated extents differed by a median of 13.9%. At

peak flood extent, this corresponds with a median expansion of the flood boundary by 0.92 m, compared to 10.9 m for the May 14 event. Differences in maximum flood depth were similarly small, and equivalent to the differences in $WSE_{90}$, differing by approximately 1%. These small differences reinforce the interpretation of fully connected floodwaters spanning Sites A and B during the July event, with consistent water surface elevations driving coherent and symmetric flood propagation from either location.

**3.3 Flood model comparison**

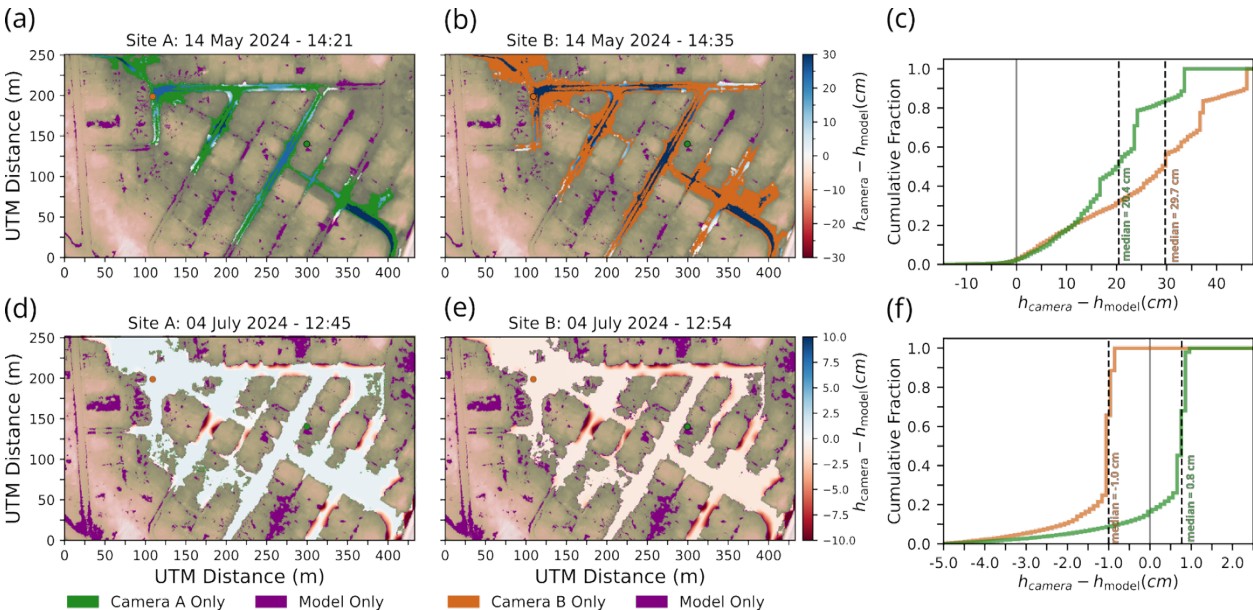

**Figure 7: Comparison between peak $WSE_{90}$ based flood extents and HEC-RAS modeled extents for the moderate case study at (a) Site A and (b) Site B and (d, e) severe case study at Site A and Site B, respectively. (c, f) Cumulative distribution of depth differences in overlapping regions for the moderate case study and the severe case study, respectively.**

Benchmarking the hydrodynamic flood model against flood-fill results shows generally good agreement in both the progression and extent of flooding, particularly during the July 4 event. The model successfully captures the broad dynamics





of inundation, though key differences emerge in the spatial structure and timing of connectivity between flood patches. During the May 14 event, the model produces disconnected flood patches, even at peak flooding, qualitatively consistent with observations at Site A (Fig 7a). However, the model does not capture the short period of connectivity estimated by flood-fill

propagation from both Sites A and B (Figure 7b). Of the modeled May 14 flood extent, eleven patches exceeded 100 m², accounting for 42% of the total modeled inundated area ($A_{model}$=0.7 ×10⁴ m²), suggesting a bias toward small, isolated flood zones. Agreement metrics for the May 14 event reflect this fragmentation (Figure 7a, b): comparing the peak flood-fill extent based on the Site A $WSE_{90}$, $F_{overlap}$=0.24, with modeled extent 43% lower than the flood-fill. At Site B the total agreement between the peak flood-fill extent was similar *($F_{overlap}$=0.21)*, with modeled extent 63% lower in area. Limiting the comparison

to the flood-fill extent within the camera FOV slightly increases agreement at Site A to $F_{overlap}$=0.34 at Site A and $F_{overlap}$=0.26 at Site B. Disagreement during the moderate event may be driven by model under-prediction. The modeled flood extent captures 32% of the extent estimated using the flood-fill procedure propagated from Site A (36% within the FOV). In contrast, 49% of the Site A-derived flood-fill extent is captured by the modeled extent (87% within the FOV). Similarly, at Site B the modeled extent covers 24% (36% within the FOV) of the Site B-derived flood-fill extent, while the Site B flood-fill extent

covers 64% (87% within the FOV) of the modeled flood extent. These spatial mismatches are accompanied by consistent underestimation in water surface elevation, with median modeled values 22 cm and 25 cm below $WSE_{90}$ at Sites A and B, respectively. Depth difference maps highlight these discrepancies, with distinct peaks aligning with the isolated modeled patches (Figure 7c).

In contrast, the estimated flood extents from the rain-on-grid model and our new method demonstrate significantly

closer agreement for the more severe, July 4 event. At the peak of camera observed flooding, the model predicts a single contiguous flood patch, accounting for 80% of the total modeled inundated area ($A_{model}$=3.86 ×10⁴ m²) and connecting Sites A and B (Figure 7d, e). Flood-fill-model agreement was also significantly higher for the July 4 event with $F_{overlap}$=0.79 and 0.77 for Sites A and B, respectively. Restricting the comparison to the within the camera FOV increases agreement to $F_{overlap}$=0.90 and 0.96 at Sites A and B, respectively. Focusing solely on the main flood patch further improves overlap to $F_{overlap}$=0.93 at

Site A and $F_{overlap}$=0.96 at Site B. Aside from minor edge effects, the model reproduces a nearly flat water surface, with median water surface elevations of 129.29 m – just 1 cm below $WSE_{90}$ at Site A and 1 cm above at Site B and yielding closely matched depth distributions (Figure 7f).

## 4 Discussion and conclusions

Urban pluvial flooding is inherently shaped by subtle variations in topography, the distribution of impervious

surfaces, and the configuration and performance of drainage infrastructure that regulates the spatial connectivity of floodwaters. These highly variable inundation dynamics can arise over very short distances and timescales, making them difficult to observe with traditional monitoring approaches. Our study demonstrated the unique strength of ground-based, time-lapse images co-registered to high-resolution topography to accurately capture these dynamics. Unlike point-based



sensors or remote sensing approaches, our method directly records the spatial and temporal evolution of floodwaters, enabling
high-resolution observation of disconnected, topographically-driven inundation patterns common in urban landscapes. By
pairing prompted image segmentation with direct topography-to-image projection, we achieved centimeter-scale estimates of
water surface elevation and time-resolved flood extents without requiring site-specific model training or in-field water-level
sensors. This allowed us to quantify spatial disconnectedness during moderate and severe storm events, track changes in flood
connectivity across topographic thresholds, and validate model predictions with an empirical, spatially explicit reference.

A major advantage of our workflow lies in the modularity of our processing pipeline and the relative ease of camera
deployment. Our use of SegmentAnything, a foundation segmentation model, allowed us to bypass the time-consuming step
of domain-specific model training, thereby accelerating flood mapping across multiple sites and events. Consistent with prior
work (Moghimi et al., 2024, Wang et al., 2024), we found that SegmentAnything performed robustly for floodwater
segmentation, producing masks with mean IoU>90% in most cases. While prompt-based segmentation has limitations for
generalization across radically different scenes (Zamboni et al., 2025), our new analysis workflow is agnostic to the specific
segmentation model used. Future applications could easily incorporate improved segmentation techniques, either domain-
specific or fine-tuned foundation models (e.g., Wagner et al., 2023), without changing the overall processing pipeline.

We found that the Static-Observer Flood Index (*SOFI*) provides a valuable proxy for site-specific flood dynamics
inferred from an image time series, particularly during large flood events. *SOFI* values tracked both the qualitative progression
of flooding and the image-derived WSE curves. In particular, during the July 4 severe event, *SOFI* rose and fell in tandem
with camera-derived hydrographs at both sites, accurately capturing the timing and magnitude of inundation. However, *SOFI*
has important limitations that constrain its broader interpretability. Because *SOFI* is calculated as the fraction of visible
inundated area within the camera field of view, its utility is highly dependent on camera pose, viewing angle, and scene
geometry. For example, *SOFI* at Site A reached a maximum value once floodwaters reached the edges of the frame, making
the metric insensitive to further increases in water surface elevation (de Vitry et al., 2019). At Site B, *SOFI* also plateaued
during ongoing inundation, as floodwaters extended away from the camera, decreasing pixel resolution and reducing the ability
to resolve further changes in water level. This behavior limits the comparability of *SOFI* values across locations or events.
Accordingly, *SOFI* should be interpreted primarily as a scene-specific indicator of relative change in water levels over time,
rather than a measure of absolute flood magnitude or spatial extent.

We show that direct topography-to-image projection provides a robust basis to enable comparison of absolute
floodwater levels between monitoring sites. By leveraging pre-existing features visible in the camera scenes, such as curbs,
streetlight poles, and driveway edges, we were able to estimate camera pose for each event without the need for permanent
ground control points or additional field surveys, even in cases where the camera had shifted slightly between floods. For our
semi-permanent camera mounts, changes in pose of up to 1° introduced approximately 5 cm of error in absolute WSE, though
relative changes within each event remained internally consistent. In contexts requiring higher precision, the use of
continuously visible, permanent ground control points (Erfani et al., 2023), fixed-mount cameras (Wang et al., 2024), or
onboard inertial measurement units (IMUs) could reduce these uncertainties. Additionally, future implementations could





integrate automated drift correction or recent advances in machine-learning-based image-to-point cloud registration (Bai et al., 2024; Jeon and Seo, 2022).

A key feature of our method is that the flood boundary identified in each image corresponds to a range of elevations, rather than a single value, likely due to slight variations in topography and minor image segmentation noise. This requires the user to select a representative elevation percentile to define the water surface elevation (WSE) for each image. In this study, we used both the 90th and 95th percentiles ($WSE_{90}$ and $WSE_{95}$) to characterize floodwater levels. Across both sites and flood events, the typical difference between $WSE_{90}$ and $WSE_{95}$ was under 5 cm, indicating consistent precision of our method within

individual flood stages. Larger differences of up to 13 to 19 cm occurred at the very beginning and end of each flood when shallow water and fine-scale topographic noise (e.g., from curb shadows or irregular pavement surfaces) introduced greater uncertainty. These uncertainties diminished as rising floodwaters filled local topography, producing smoother and more stable elevation distributions at the floodwater edges. Despite these sources of uncertainty, our results show that extracted WSE values closely match observed inundation timing. In particular, the consistent rise and fall of WSE during the July event, most

notably during the rising limb, further confirms the method's ability to resolve spatial flood dynamics at scales and frequencies that conventional sensors cannot achieve.

      The contrast in water surface elevation (WSE) dynamics between the May 14 and July 4 flood events illustrates how our method captures spatial flood connectivity in urban landscapes. During the moderate May 14 event, $WSE_{90}$ time series at Sites A and B revealed distinct, asynchronous flood pulses, with peak elevations differing by up to 16.2 cm. Site A exhibited

two short-lived pulses separated by dry conditions ($SOFI$=0), while Site B experienced a more continuous rise and fall in water level. These discrepancies support the interpretation that floodwaters occupied disconnected topographic depressions, each filling and draining independently in response to localized rainfall, infiltration, and drainage behavior. Our results and interpretation are consistent with similar methods that have been successfully applied to studying connectivity and surface-water flow between natural wetland depressions (McLaughlin et al. 2019). In contrast, during the more severe July 4 event,

WSE time series at both sites rose and fell in tandem, with peak elevations differing by just 1.7 cm – well within the range of measurement uncertainty. This tight correspondence in timing and magnitude of water level changes strongly indicates persistent hydraulic connectivity throughout the event. The high agreement between cameras across the full hydrograph demonstrates the value of our dual-camera system in confirming both the onset and spatial extent of flooding and in identifying when and where discrete flood patches transition into a single, continuous water surface.




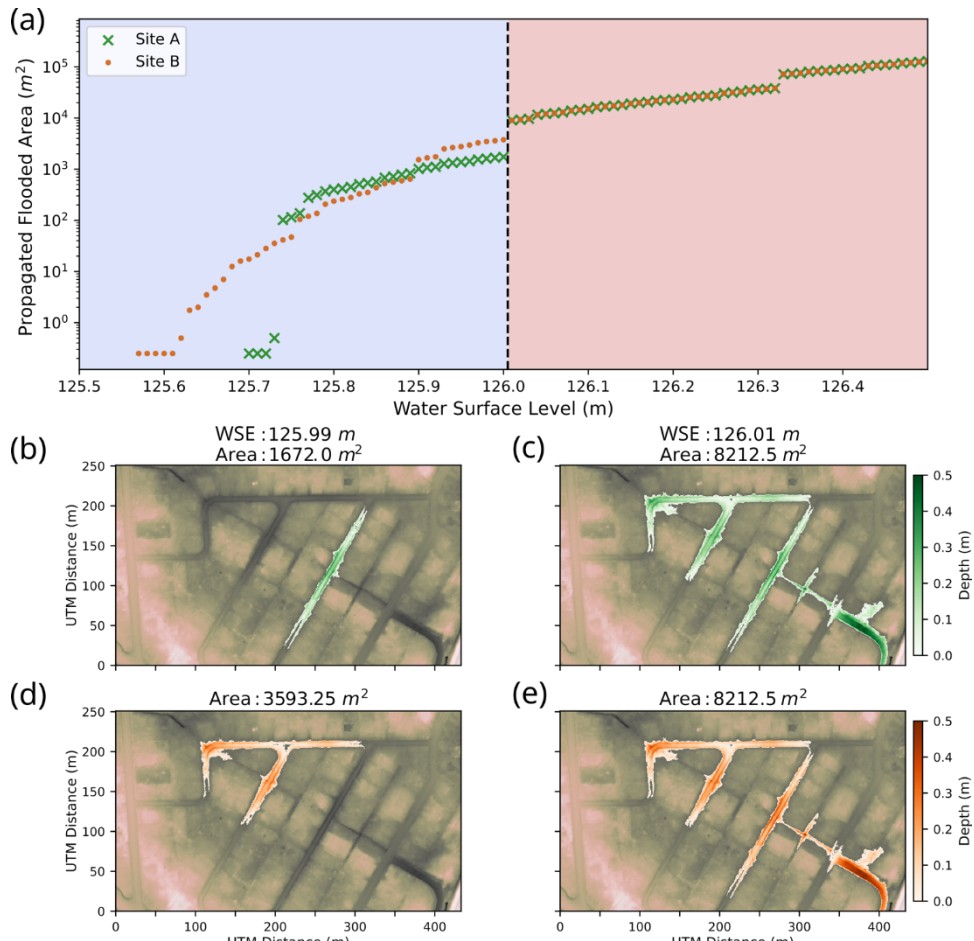

**Figure 8: (a) Flood-fill-estimated inundated area as a function of $WSE_{90}$ for Sites A (green) and B (orange) with threshold elevation denoted with a dashed line, (b) Site A-derived flood-fill extent for $WSE_{90}$ =125.99 m and (c) $WSE_{90}$ = 126.01 m, and (d) Site B-derived flood-fill extent for $WSE_{90}$=125.99 m and (e) $WSE_{90}$=126.01 m**

Our flood-fill propagation method, guided by WSEs estimated using our topography-to-image projection pipeline, provide a useful basis for assessing the spatial extent of floodwaters beyond the image frame. Flood-fill approaches offer a relatively simple, computationally efficient method for exploring surface water connectivity across complex urban topography. In both natural and engineered landscapes, surface water connectivity is governed by the size, depth, and arrangement of topographic depressions and their associated spill point thresholds, which control the extent of flooding (Leibowitz et al., 2016; Samela et

al., 2020; Maksimovic et al., 2009; Lee et al., 2023). Like other "bathtub models", this approach assumes zero flow resistance and instantaneous water propagation, leading to highly non-linear relationships between water surface elevation and inundated flood area (Sanders et al., 2024). In our study area, flood-fill simulations propagated from Sites A and B reveal distinct connectivity patterns, with abrupt jumps in flooded extent when water surface elevations (WSE) reach key topographic spill points (Figure 6a). The most prominent of these occurs at ~126.01 m, where just a 2 cm increase in WSE, from 125.99 m

(Figure 8b,d) to 126.01 m (Figure 8c, e), results in a 390% increase in flood area at Site A and 270% at Site B. Above this



threshold, water surface elevation increases at both sites is identical, indicating that the two sites are fully hydraulically connected.

However, our results from the moderate May 14 flood event reveal key limitations of the flood-fill approach for representing urban flood dynamics. Water surface elevations derived from our topography-to-image projection show a consistent ~10 cm difference between Sites A and B. This magnitude of difference, over only a few hundred meters, provides direct, empirical evidence that the flood patches remained poorly connected, or fully disconnected, throughout the event. Despite this, flood-fill extents predicted full connectivity, as water levels at both sites exceeded the flood-fill model's 126.01 m threshold, including during the observed gap in flooding at Site A. This mismatch illustrates how purely elevation-based models can overestimate floodwater connectivity by neglecting important factors such as microtopographic barriers that cannot be resolved by the 0.5 meter-resolution DEM, infiltration losses, and stormwater infrastructure that can disrupt surface flow even when spill thresholds are surpassed (Lee et al., 2023; Shrestha et al., 2022). The sensitivity of flood-fill results to DEM characteristics further limits their reliability for predictive applications. In our case, switching from a TIN to an IDW-interpolated DEM increased the flood-fill-predicted elevation threshold by 4 cm, illustrating how changes in DEM processing method and resolution alone can shift the timing and extent of predicted connectivity (de Almeida et al., 2016). In contrast, our camera-derived WSE estimates directly capture the spatial and temporal behavior of floodwaters, revealing asynchronous dynamics and persistent disconnection that would otherwise be invisible to single-point or flood-fill only approaches. This May 14, moderate flood event highlights that while flood-fill approaches remain useful for rapid flood assessment (Preisser et al., 2022), high-resolution, empirical observations are required to better constrain, validate, and improve models of flood connectivity in urban landscapes.

Our results highlight the significant potential of image-based flood monitoring to improve calibration and evaluation of urban flood models, particularly physics-based hydraulic modeling approaches (de Vitry and Leitão, 2020). While depression-based models with volume accounting and simplified inclusion of drainage systems (e.g., Maksimovic et al., 2009; Samela et al., 2020) offer a more realistic alternative to simple flood-fill, fully 2D hydrodynamic models remain the benchmark for predictive urban flood forecasting (Guo et al., 2021; Rosenzweig et al., 2021). Recent advances in urban flood modeling have expanded the capabilities of 2D hydrodynamic models through features such as Rain-on-Grid water input and coupling with 1D sewer-stormwater systems in commonly used software packages like HEC-RAS (Sañudo et al., 2020; Guo et al., 2021). These developments have enabled more realistic simulation of complex, infrastructure-mediated flood behavior in urban settings, accounting for both overland flow and subsurface drainage. However, the utility of these models remains limited by the availability of empirical calibration data, especially for localized pluvial events where traditional stream gauges are absent. Engineered drainage can dramatically alter flood response, with Anni et al. (2020) finding up to a 20-fold increase in modeled flood volume when stormwater losses are not included. This sensitivity is further amplified in urban areas with aging or neglected infrastructure, where drainage performance may vary over time (Shrestha et al., 2022).

In such settings, camera-derived WSEs offer a rare empirical reference for validating modeled water levels and spatiotemporal patterns of inundation. These high-resolution, time-resolved observations enabled direct comparison with



outputs from an uncalibrated HEC-RAS Rain-on-Grid simulation of the July 4 flood event. The close match between observed and modeled peak flood depth, timing, and extent demonstrates the strong potential of integrating image-derived data into calibration workflows for 2D hydrodynamic models. This proof-of-concept highlights the value of our method in high-flow scenarios where floodwaters are hydraulically connected and drainage networks are overwhelmed, offering a practical, data-driven way to constrain uncertainty in urban flood simulations. Beyond event reconstruction, these observations can support

applications such as real-time model updating, performance evaluation of stormwater infrastructure, and planning for flood mitigation in poorly instrumented or rapidly evolving urban settings.

In contrast, for the more moderate May 14 event, the model underpredicted total flood extent. These discrepancies may reflect known challenges in simulating shallow, spatially variable flooding, where results are highly sensitive to initial conditions, roughness parameters, and the representation of drainage behavior (de Almeida et al., 2018). In our case, they also

stem from limitations in the flood-fill-based propagation used for comparison, which overestimated surface connectivity due to DEM resolution constraints and lack of drainage detail. The mismatch between predicted and observed connectivity for this smaller event illustrates how subtle differences in topography, infiltration, or active drainage (e.g., pumping) can lead to large differences in modeled flood behavior. For example, human interventions such as pumping by the utility truck at Site B during our moderate flood event are immediately apparent in camera images and may give context to the rate of flood recession that

would be absent from rain-on-grid model output, or pressure-based water level loggers.

Despite these challenges, our results demonstrate how empirically-derived WSEs can complement and strengthen traditional hydraulic modeling workflows. Our method provides continuous, high-resolution estimates of water level and extent that are directly tied to real flood behavior, capturing sub-decimeter changes in WSE and floodwater connectivity that would otherwise be missed by more traditional flood monitoring and modeling approaches. These observations are especially valuable

for model calibration in settings with no gauges or rapidly changing infrastructure performance. As stormwater systems become increasingly strained by climate extremes, integrating data-driven camera networks with physically-based modeling frameworks offers a promising pathway for improving urban flood forecasting, response, and planning.

The need for actionable urban flood data is greatest in underserved communities where existing monitoring is limited, and deficiencies in large-scale flood-risk assessments often go unnoticed (Schubert et al. 2024). Closing these gaps in flood risk

assessment requires empirical flood observations at scales ranging from individual streets to specific properties. Our project, conducted in coordination with Cahokia Heights residents, offers a practical solution towards the development, deployment and operation of a low cost, camera-based flood monitoring system. The design of any community-based monitoring project must balance both technical requirements and measures to protect the privacy of and minimize intrusiveness to residents (de Vitry et al., 2019; Aziz et al. 2023). On this front, low-cost non-contact sensors can be an ideal solution for scalable flood

observation in an urban environment (Mydlarz et al., 2023). The camera stations used in this project can be constructed for approximately $100 USD and installed in approximately 15 minutes, allowing for the relatively rapid deployment of large networks. A major advantage of semi-permanent cameras compared to other sources of flood images, such as public webcams, security cameras, or crowdsourced photos, is the flexibility to adapt the network while otherwise maintaining stability in



observations (Helmrich et al., 2021). Based on both our own observations and resident feedback, camera position and image
settings can be readily adjusted to iteratively improve the quality of flood observations. Beyond scientific data collection,
community-focused monitoring also has an important role to play in the communication of flood risk and impacts (Mydlarz et
al. 2023). Specifically, visual images of street-level flooding provide a tangible and easily interpretable data product for non-
specialists compared to traditional products such as flood-frequency maps (Siegel and Kulp, 2021).  To this end, camera-based
flood observations can both fill critical data gaps related to urban nuisance flooding and provide communities with direct,
actionable insights into the frequency and severity of pluvial flooding.

In this contribution, we present a novel, camera-based approach to urban flood monitoring that integrates time-lapse
imagery with high-resolution topography to estimate water surface elevation and flood extent with centimeter-scale precision.
Our method offers a flexible, low-cost solution for capturing urban flood dynamics, capturing highly localized events that are
difficult to monitor with conventional tools.  By combining foundation segmentation models with direct topography-to-image
projection, we bypass the need for in-field water-level sensors and site-specific model training, enabling rapid deployment and
scalability across sites. Our observations not only captured asynchronous flood dynamics and topographically driven
differences in flood connectivity during moderate and severe flood events but also provided a rare empirical dataset for flood
model validation. Comparisons with flood-fill and 2D hydrodynamic models showed varied success in reproducing observed
flood behavior, highlighting the potential of our method to improve pluvial urban flood representation in risk assessments.
Moving forward, this approach can enhance urban flood resilience by enabling real-time monitoring and more accurate
forecasting to support emergency response and infrastructure planning. Additionally, integration of camera-based monitoring
with hydrodynamic flood models can close critical data gaps in urban hydrology, improving understanding and management
of complex flood processes in urban landscapes.

**Data Availability:**

The aerial lidar data for St. Clair county used in the study is available through the Illinois Geospatial Data Clearing House
(https://clearinghouse.isgs.illinois.edu/data/elevation/illinois-height-modernization-ilhmp)              or              OpenTopography
(https://portal.opentopography.org/usgsDataset?dsid=IL_HicksDome_FluorsparDistrict_B1_2019). The precipitation data
used in the model is available through MesoWest (https://mesowest.utah.edu/).  To protect the privacy of community residents,
georeferenced flood extent data and raw imagery are not publicly available.

**Code Availability:**

The code used for image processing, flood extent propagation, and model comparison, is available via Zenodo at
https://doi.org/10.5281/zenodo.16414887

*A Note to Editors and Reviewers:  All data are publicly availably on Zenodo as a draft, however, the data are not yet formally
published with a DOI. The formally published data will be cited here and linked with a DOI following review. This delay is to*

*enable edits if substantive methodological changes are suggested during the review process resulting in material changes to*
*the assets in the current Zenodo draft repository. A link with access to the draft repository is given below.*
*https://zenodo.org/records/16414887?preview=1&token=eyJhbGciOiJIUzUxMiJ9.eyJpZCI6IjAzM2MxZjBhLTJjNWEtNDI1*
*MS04ZWE1LTRlODJlZTkzMjEyNCIsImRhdGEiOnt9LCJyYW5kb20iOiJhNmVlNDQ0Y2Y0Njc3MTFiZDQ0MzAzMGI5ZDF*
*mYmNkOSJ9.fNd50BKqMzWA7NBgVwrWqpGVKyLTJFSjcn_yawfLlnX3YDGyoL1NwX4-qnnuGKgT6coHGLrntXJGKay-*
*RhatKw*

**Competing interests:**

The contact author has declared that none of the authors has any competing interests.

**Financial support:**

This material is based upon work supported by the National Aeronautics and Space Administration under Grant No.
80NSSC22K1653 to CCM, the National Science Foundation under Award No. 2026780 to CCM and Award No. 2026789 to
JAC, and a seed grant from the Transdisciplinary Institute in Applied Data Sciences at Washington University in St. Louis.

**Author contributions:**

CCM and JAC formulated the study scope. JED, SD and CCM installed and maintained the camera stations. JED conducted
the GNSS and lidar surveys. JED developed the image analysis methods and completed data analysis with contributions from
SD and guidance from CCM. SD developed the HEC-RAS model with guidance from CCM. JED and CCM wrote the
manuscript with input from SD and JAC.

**Acknowledgments:**

Lidar and GNSS equipment were provided by the Fossett Laboratory for Virtual Planetary Exploration at Washington
University in St. Louis. Multiple undergraduate and graduate students assisted with fieldwork, including Henry Chandler,
Valencia Ajeh, Robert Kostynick, Isabel Lopez, and Cesar Lopez. Hossein Hosseiny contributed to preliminary flood model
development. Finally, we thank the Cahokia Heights residents who volunteered to have camera stations installed on their
property for ongoing support and collaboration of flood monitoring efforts.

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
