# Peer review of "Community-scale urban flood monitoring through fusion of timelapse imagery, terrestrial lidar, and remote sensing data"

_EGUsphere, 2025_

## Author Comment (AC2)

The paper is very interesting and presents an application of camera-based flood monitoring. I do not fully agree with some of the terminology used to describe existing approaches, particularly the use of the term "traditional," but this does not significantly affect the main message of the paper. Given the rapid improvement in machine learning methods, camera quality, and the continuously decreasing costs of imaging systems, I see strong potential for camera-based approaches in the near future. The proposed methodology is timely and well aligned with these developments. Overall, I think the paper can be considered for publication after the authors address the comments listed below and other reviewers' comments.

We thank the reviewer for their thoughtful comments on our manuscript. We answer any questions and describe any changes made to the text below each comment below.

1- The statement that traditional fluvial monitoring infrastructure (e.g., stream gauges and water level sensors) is not suited to detect spatially disconnected pluvial flood patches appears overstated. The manuscript should acknowledge that recent advances in dense sensor networks, smart drainage monitoring, and urban hydrometric instrumentation partially address these limitations. The authors are encouraged to moderate the claim and more clearly delineate the *specific contexts* (e.g., highly localized, shallow, short-duration inundation) in which conventional monitoring remains insufficient.

2- Lines ~(75–90): The manuscript currently frames alternative approaches in a way that may be interpreted as dismissive of prior work. It is not necessary to portray existing methods as fundamentally inadequate to justify the proposed approach. A more balanced framing that highlights complementary strengths and weaknesses particularly regarding accuracy, spatial resolution, temporal resolution, and operational constraints would strengthen the motivation and credibility of the study.

We thank the reviewer for these comments. We have added additional context on other state of the art monitoring techniques to better highlight the specific applications where camera-based monitoring has advantages, both in terms of flooding process, and implementation/operational considerations (Mydlarz et al. 2024; Silverman et al. 2024; Gold et al 2023). We have also modified the language to avoid appearing dismissive of other approaches, but still highlight where camera-based approaches can build new capacity.

*Lines 81-86:* "*Both contact sensors (e.g., pressure transducers) and non-contact sensors (e.g., radar, ultrasonic), have proven effective for monitoring distributed urban flooding (Mydlarz et al. 2024; Gold et al. 2023). However, they can face operational challenges for small-scale flooding in urban settings, including limited installation locations and sensitivity to local disturbances (Song et al., 2024). For example, the radar based FloodNet system was limited at many sites to placement over sidewalks, limiting observation of early road flooding (Mydlarz et al. 2024).* "

*Lines: 798-821:* "*Cameras offer a flexible, low-cost, and highly informative option for distributed monitoring in settings where flooding dynamics are poorly understood. However, even under ideal conditions image-based water level estimates are unlikely to reach the absolute accuracy of pressure-based*

*sensors, and the sensitivity of image quality to environmental conditions make them less well suited for contexts where the consistency of measurement in paramount. Particularly when more direct public sector cooperation is possible, storm drain installed pressure sensors have proven highly valuable for realtime distributed flood monitoring (Gold et al. 2024; Silverman et al. 2024). However, even in these cases, cameras can serve as important complement to non-visual sensors. Gold et al. 2024 used co-located cameras and storm-drain pressure sensors, with images helping to identify cases where storm drain based measurements may be impaired. Another major advantage of semi-permanent cameras compared to other sources of flood images, such as public webcams, security cameras, or crowdsourced photos, is the flexibility to adapt the network while otherwise maintaining stability in observations (Helmrich et al., 2021). While camera sensors themselves are highly available, the requirement of high-resolution topographic data can still be a barrier to broader application of our methods. However, advancements such as smartphone mounted lidar, and national scale datasets like 3DEP have reduced this. Future research could leverage this framework to optimize camera network configuration, balancing the number and placement of ground-based cameras to maximize spatial coverage and the ability to observe flood connectivity (Negri et al., 2025; Zhao et al., 2025)."*

3- The challenge of translating two-dimensional image-based flood fractions into real-world water depth should be explained more rigorously. In heterogeneous urban environments, flooded pixel fraction does not scale linearly with water depth because inundation often occurs in shallow, spatially discontinuous depressions controlled by microtopography, curbs, and drainage infrastructure. Small vertical changes in water level can produce large apparent changes in flooded area (or vice versa), leading to ambiguity when inferring depth or volume from image coverage alone. Explicitly linking this limitation to urban surface complexity would clarify why image-only approaches are insufficient for depth estimation.

In the revised discussion we draw more explicit contrast between the image-only SOFI characterization of flooding, and the 3D-2D projection-based approach. Inundated area shows a nonlinear increase in the presence of depressions (Figure 7), with further bias in image coverage produced by camera scene geometry (SI Figure 3). While still subject to inherent resolution and visibility limitations, 3D-2D projection can overcome this limitation by explicitly accounting for the relative geometry of flood extents to the camera:

*Lines 258-262: "This ratio is referred to as the Static Observer Flooding Index (SOFI), following the approach of Vitry et al. (2019), providing a simple proxy for flood intensity as seen from a fixed observation point. SOFI has been shown to correlate strongly with changes in water level for a given location (Moy de Vitry et al. 2019). The shape and magnitude of SOFI response depend strongly on the geometry of a camera relative flooding, and as such values cannot be directly compared between study sites."*

*Lines 563-566: "Accordingly, SOFI should be interpreted primarily as a scene-specific indicator of relative change in water levels over time, rather than a measure of absolute flood magnitude or spatial extent."*

4-Use of the term "traditional" The repeated use of the term *traditional* to describe existing monitoring and modeling approaches is potentially misleading. Many of these methods are actively evolving and increasingly integrated with high-resolution data and advanced numerical schemes. The authors may consider replacing this term with more precise language to avoid implying obsolescence.

In revision we will remove use of the term 'traditional' throughout and instead refer to either specific technologies or specific distinguishing features, such as between contact/point-based and non-contact/continuous methods.

5-Is this method having a lower cost in compare with existing approaches? this argument is weakened by reliance on aerial LiDAR and high-density terrestrial LiDAR. Why the authors should not explicitly discuss the cost, accessibility, and transferability of these datasets, particularly for low-income or data-scarce regions. A comparative table summarising data requirements, costs, spatial/temporal resolution, and uncertainties across camera-based methods, LiDAR-dependent approaches, and conventional monitoring would provide a transparent and unbiased comparison.

We have moderated the language to instead discuss cameras as one member of a suite of distributed, low-cost monitoring tools. However, we emphasize that the aim of this study is not a ready-made analysis package or hardware platform and feel that a specific cost-breakdown comparison of our deployment may not be representative of camera monitoring as a whole. And given the significant project and site-specific factors involved in sensor network cost feel that explicit cost comparison would be a diversion from the main goal of demonstrating our methodology. We have however added discussion of tradeoffs between resolution, accuracy and cost/flexibility between methods. We recognize that there are still material and data barriers to applying some of our methods. However, we note that most elements of our workflow could be replicated in less intensive ways, albeit with scale and accuracy tradeoffs. For example, at the scale of an individual camera recent studies have leveraged smartphone integrated lidar and photogrammetry for water level prediction (Erfani et al. 2023), while the aerial lidar was freely obtained from the USGS 3DEP program. (Refer to Lines 726-739 in our reply to Questions 1 & 2)

6-The workflow for estimating floodwater elevation is central to the contribution of this paper, yet it is difficult to fully evaluate due to incomplete access to it (the Zenodo link could not be accessed, at least I could not).

The current repository is still in 'draft' mode to facilitate any material additions or changes prompted by the review process. The sharing link in the Code Availability section (reproduced below) should give access and we are happy to generate and share a new link if needed.

7-The Discussion and Conclusions section is lengthy and combines interpretation with summary statements. I recommend separating this into a concise Discussion section focused on interpretation and limitations, followed by a distinct Conclusions section that succinctly highlights the main contributions, findings, and implications.

We will add subheadings to the discussion and separate the conclusions into a separate and final section.

8-I understand the study is not centred on HEC-RAS, modelling, however, providing a brief description of the HEC-RAS setup, assumptions, and any calibration or validation strategy (ideally in supplementary material) would strengthen the credibility of the comparison without disrupting the narrative flow of the main manuscript.

We have migrated details regarding HEC-RAS model design and execution from the code supplement into the main text and supplement (See Supplementary Text 1). We have also added additional explanation of the motivation, and limitations of the current model comparison:

*Lines 372-375: "Because the model itself is only qualitatively calibrated, its output is not treated as a direct validation for absolute water levels estimated from images. Instead, it characterizes similarity or divergence in flood behavior predicted by each method, based. This is quantified both in terms of the relative agreement in predicted flood extent, and spatial flood connectivity, between the two methods."*

*Lines 707-715: "Despite these challenges, our results demonstrate how empirically-derived WSEs can complement and strengthen traditional hydraulic modeling workflows. Our method provides continuous, high-resolution estimates of water level and extent that are directly tied to real flood behavior, capturing sub-decimeter changes in WSE and floodwater connectivity that would otherwise be missed by point-based flood monitoring and modeling approaches. While further validation of camera-derived extents would be necessary for confident direct calibration, this level of precision is valuable for the initial validation of uncalibrated models, an important tool for preliminary flood-risk analysis in settings with no gauges or rapidly changing infrastructure performance. As stormwater systems become increasingly strained by climate extremes, integrating data-driven camera networks with physically-based modeling frameworks offers a promising pathway for improving urban flood forecasting, response, and planning. "*

9-The manuscript would benefit from a brief discussion of how segmentation uncertainty propagates into water surface elevation estimates, particularly under challenging conditions such as specular reflections, shadows, low-light conditions, and partial occlusions by vegetation or vehicles. (At least raise them)

This point is well taken and an important aspect of the broader viability of these methods. This is partly included in the discussion of trade-offs of camera-based monitoring in our response to

question two. We will also add discussion of both random and systematic segmentation error, noting the ways that our interpolation approach partly reduces their influence:

*Lines 709-723:*

*"4.4 Accuracy and error propagation*

*Elements of our method including the spatial aggregation of flood boundaries, and the calculation of multiple water level thresholds effectively limit these uncertainties to a magnitude sufficient for urban flood characterization. Water level estimates are robust to both minor random and systematic error in flood mask segmentation. For the severe event at Site A, water levels calculated separately for the half of the scene with a parked park car partly obscuring the water line, differed by a mean of less than 2 cm from water levels. For the same event, introducing random jitter of 10 to 20 pixels to the flood boundary similarly resulted in a mean water level difference of less than 2 cm. More severe errors in flood segmentation or camera pose that are not mitigated are detectable in artifacts such as large re-projection errors, asymmetry in the projected flood extents, or exaggerated ranges between $WSE_{90}$ and $WSE_{95}$. Additionally, the visual context provided by images allows for qualitative evaluation of agreement with visual flood markers such as road over-topping. In addition to water level uncertainty, flood extent estimates are influenced by the quality of topographic data. Even with robust georeferencing, both physical landscape change between data collections, and artifacts from differences in point density will produce localized elevation differences. In flood extents propagated on the aerial lidar are biased towards over-prediction, due to limited representation of fine scale topographic structure. Both water level estimation and flood extent propagation are more sensitive at lower water levels. As such, interpretations of discreet changes in flood connectivity from small water level increases should be qualified. Future work should further explore the propagation of error between these sources."*

Reply Citations:

Hong, Y., Kessler, J., Titze, D., Yang, Q., Shen, X., & Anderson, E. J. (2024). Towards efficient coastal flood modeling: A comparative assessment of bathtub, extended hydrodynamic, and total water level approaches. Ocean Dynamics, 74(5), 391-405.

Gallien, T. W., Sanders, B. F., & Flick, R. E. (2014). Urban coastal flood prediction: Integrating wave overtopping, flood defenses and drainage. Coastal Engineering, 91, 18-28.

Li, Z., Mount, J., & Demir, I. (2022). Accounting for uncertainty in real-time flood inundation mapping using HAND model: Iowa case study. Natural Hazards, 112(1), 977-1004.

Silverman, A. I., Brain, T., Branco, B., sai venkat Challagonda, P., Choi, P., Fischman, R., ... & Toledo-Crow, R. (2022). Making waves: Uses of real-time, hyperlocal flood sensor data for emergency management, resiliency planning, and flood impact mitigation. *Water Research*, *220*, 118648.

Gold, A., Anarde, K., Grimley, L., Neve, R., Srebnik, E. R., Thelen, T., ... & Hino, M. (2023). Data from the drain: A sensor framework that captures multiple drivers of chronic coastal floods. Water Resources Research, 59(4), e2022WR032392.

Negri, R., Ceferino, L., & Cremen, G. (2025). Prioritizing urban areas for the deployment of hyperlocal flood sensors using stakeholder elicitation and risk analysis. Natural Hazards Review, 26(3), 04025020.

Zhao, Z., Liang, Y., Wang, K., Ding, X., Zhang, Y., & Hu, C. (2025). Collaborative sensing optimization layout model of heterogeneous sensors under urban flooding environment. Journal of Hydrology, 650, 132528.

---

## Author Comment (AC3)

RC2: 'Comment on egusphere-2025-3962', Anonymous Referee #2, 11 Dec 2025

In their paper, the authors present an innovative urban flood monitoring approach. Intersecting segmented flood masks derived from imagery recorded by low-cost trail cameras and lidar data, they estimate flood water surface elevations for two flood events. Maximum flood depths and extents were then compared with results from a 2D hydrodynamic model.

I have read the paper with interest and think it can be published after major revision. My detailed comments are included below.

Major concerns

1. HEC-RAS model

While I agree in general that comparing flood extent and/or depth from the authors' new method with results from a 2D hydrodynamic model might be an interesting analysis, the paper in its current for lacks important details regarding how the HEC-RAS model was implemented:

*[Q1]* The details of the HEC-RAS model implementation are currently included in the Zenodo data supplement. Because the HEC-RAS model itself is secondary to the development of our analysis pipeline, we do not want to overwhelm the readers with too many details in the main text given the current length of the manuscript. We will migrate the relevant elements into the main text methods section and supplementary information.

How was the model grid set up?

We have added the following to the supplement:

*(Text S1) Model domain: The computational mesh was generated from a TIN-interpolated, and gap filled, 0.5 m resolution DTM generated from 2019 USGS 3DEP aerial lidar data. The base mesh was generated with 10 m node spacing. Breaklines were added for channel centerlines, culvert inflows and storm drains. Mesh refinement was applied within 5 m of these breaklines, reducing node spacing to 1 m for the area of interest.*

What infiltration method was used in the rain-on-grid approach, and how was it parameterized? What land use classifications and corresponding roughness coefficient were used?

We have added the following to the supplement:

*(Text S1) Mannings roughness (n) and runoff: 30 m resolution National Land Cover Database (NLCD) were used to define spatially variable roughness coefficients, using HEC-RAS manual reference values for each classification. This was refined using vector polygons of road surfaces and building footprints from the Illinois Department of Transportation. Within building footprints n was assigned a high value of 10 which prevents the routing of runoff from those cells.*

| Land Cover Classification | Manning's N Value |
|---|---|
| NoData | 0.035 |
| Roads | 0.01 |
| Buildings | 10 |
| Banks | 0.04 |
| MainChannel | 0.03 |
| Open Water | 0.035 |
| Developed, Open Space | 0.035 |
| Developed, Low Intensity | 0.08 |
| Developed, Medium Intensity | 0.12 |
| Developed, High Intensity | 0.15 |
| Barren Land Rock/Sand/Clay | 0.025 |
| Deciduous Forest | 0.15 |
| Evergreen Forest | 0.15 |
| Mixed Forest | 0.1 |
| Shrub/Scrub | 0.9 |
| Grassland/Herbaceous | 0.04 |
| Pasture/Hay | 0.045 |
| Cultivated Crops | 0.05 |
| Woody Wetlands | 0.07 |
| Emergent Herbaceous Wetlands | 0.07 |

*Table S2:  Reference landcover based Manning 's roughness coefficients taken from USACE 2024.*

**Rainfall Runoff:** *The curve number (CN) method was used to calculate initial infiltration losses and runoff generation.  The same NLCD landcover, and IDOT building and road layers were used to define spatially variable values for CN, abstraction ratio and minimum infiltration rate. Reference values were taken from the HEC-RAS hydraulics manual, with abstraction ratios suggested by Hawkins and Jiang 2023. No additional infiltration losses are calculated after the initial rainfall-runoff conversion.*

| Name
(Land Cover: Soil Hydric Group) | Curve Number | Initial Abstraction Ratio | Abstraction Ratio | Minimum Infiltration Rate |
|---|---|---|---|---|
| Developed, Medium Intensity : D | 86 | 0.05 | 0.082 | 1.270 |
| Developed, Open Space : B/D | 74 | 0.05 | 0.176 | 3.485 |

| | | | | |
|---|---|---|---|---|
| Emergent Herbaceous Wetlands : A | 76 | 0.05 | 0.158 | 7.600 |
| Grassland/Herbaceous : D | 80 | 0.05 | 0.125 | 1.270 |
| Buildings : | 100 | 0.05 | 0.000 | 0.000 |
| Roads : C | 98 | 0.05 | 0.010 | 0.000 |

Was storm drain infrastructure modeled, or only surface flow?

We have added the following to the supplement:

> *(Text S1) Stormwater system:* *The combined sewer-stormwater system was modeled as a 1D pipe network in HEC-RAS. The location of stormwater inflows were taken from the Illinois Department of Natural Resources (IDNR) and Heartlands Institute survey of the Prairie Du Pont Watershed. Precise information on the topology and hydraulics of the sewer-stormwater system is unavailable, and connection between inflows was inferred based on published IDNR and USACE reports and maps (USACE 2024; IDNR 2023). Pipe diameter of 0.7 m, based on IDNR survey, and n of 0.015 m were used, based on USACE reported values. The pipe network was connected to the know drainage ditch outfall.*

Are there storage areas within the model domain?

> Because the study area is separated from nearby lakes and reservoirs by the major drainage ditches and roads no additional storage areas were included.

Without additional detail, a review of this portion of the analysis is nearly impossible. Even if detail is added, I still question the value of comparing the authors results with those from an uncalibrated HEC-RAS model; I also suspect the lack precipitation data form within the study area adds substantial uncertainty to model results (it sounds like data from only one rain gauge was used for each event, and gauges were located at a distance of 6 and 8 km from the study area, respectively).

**[Q3]** We Thank the reviewer for their comments, but even without quantitative calibration we believe there is significant value in our model comparison. A major motivation of our study is monitoring approaches for areas without sufficient data for pluvial model calibration. This approach is not unique to our study. There are multiple prior studies where uncalibrated, and otherwise simplified 2D models are used to evaluate new DTM based methods (e.g. Samela et al. 2020; Preisser et al. 2022). To that end, the comparison is intended to identify major similarities and differences in characteristic behavior between camera-derived flood extents and a rain-on-grid model, not as an absolute ground-truth. While factors like precipitation uncertainty will modify maximum flood timing and extent, they are unlikely to alter the behaviors implicit to a rain-on-grid model which distinguish it from our image-based estimates. We will revise the methods and

discussion to clarify our conceptual approach and qualify the limits of direct camera to model validation:

> ***Lines 372-375:*** *"Because the model itself is only qualitatively calibrated, its output is not treated as a direct validation for absolute water levels estimated from images. Instead, it characterizes similarity or divergence in flood behavior predicted by each method, based. This is quantified both in terms of the relative agreement in predicted flood extent, and spatial flood connectivity, between the two methods."*

> ***Lines 707-715:*** *"Despite these challenges, our results demonstrate how empirically-derived WSEs can complement and strengthen traditional hydraulic modeling workflows. Our method provides continuous, high-resolution estimates of water level and extent that are directly tied to real flood behavior, capturing sub-decimeter changes in WSE and floodwater connectivity that would otherwise be missed by point-based flood monitoring and modeling approaches. While further validation of camera-derived extents would be necessary for confident direct calibration, this level of precision is valuable for the initial validation of uncalibrated models, an important tool for preliminary flood-risk analysis in settings with no gauges or rapidly changing infrastructure performance."*

I think one of the potentially important applications of the proposed method is mentioned in the discussion (lines 588-589): data for calibration of hydrodynamic models is limited, particularly for pluvial flooding. Here, estimates of water surface elevations and flood extents from cameras could fill an important data gap. If the authors could demonstrate that they can calibrate their HEC-RAS model using camera-based observations, that would strengthen the paper considerably.

While we agree with the reviewer that there is significant future opportunity in using camera-based observations to calibrate flood models (which we discuss in Lines 810-815), we feel that this effort is beyond the scope of the study presented here. The focus of the current study is application of the computer vision methods to estimate spatial flood extent, and introducing an additional model calibration element is likely to detract from that focus. The reviewer's suggestion would require us to simultaneously evaluate the performance of our camera-based methods while also applying those methods to calibrate the HEC-RAS model, introducing ambiguity into the interpretation of both elements of the study.

We will revise the discussion to more directly call out the potential use of these methods for future flood model calibration as follows:

> ***Lines 847-863****: "Camera-based observations provide a promising avenue to address these calibration gaps. Depending on the data available and the precision required, camera-derived information could support multiple levels of model calibration. At a minimum, observations of flood presence, extent, and connectivity can serve as semi-quantitative validation of model structure and behavior. More detailed or well-distributed camera installations could function as stream-gauge surrogates, enabling direct calibration of key model parameters such as surface roughness, stormwater capacity, or flood wave timing. These approaches could ultimately facilitate both post-event model evaluation and real-time model adjustment, bridging gaps in empirical data for urban flood forecasting. When possible to implement, camera-derived WSEs offer a rare empirical reference for validating modeled spatiotemporal patterns of inundation. For example, these high-resolution, time-resolved observations enabled direct comparison with outputs from an uncalibrated HEC-RAS Rain-on-Grid simulation of the July 4 flood event, revealing a close match in peak flood depth, timing,*

*and extent. This proof-of-concept highlights the strong potential of integrating image-derived data into calibration workflows for 2D hydrodynamic models, particularly in high-flow scenarios where floodwaters are hydraulically connected and drainage networks are overwhelmed. Beyond event reconstruction, such observations can support real-time model updating, performance evaluation of stormwater infrastructure, and planning for flood mitigation in poorly instrumented or rapidly evolving urban settings, providing a practical, data-driven way to reduce uncertainty in urban flood simulations."*

2. Extension of flood extends beyond the camera field of view

The authors apply a flood fill procedure to estimate flood extents outside the camera field of view. I question this approach, which can't account for overland flow dynamics, infiltration, etc. I think the authors might be better off using the flood fill approach only within the field of view.

**[Q5]** We thank the reviewer for their comment. However, there is an important conceptual distinction between flood-fill models and process-based flood models. By prescribing a surface water level the flood-fill extents are agnostic to the factors such as infiltration that produced the water level. Further, flood fill models do not contain an explicit time-dependence, each flood extent is independently generated from a single image observation. Within the context of pluvial flooding within an urban area, where the fraction of impervious surfaces is quite high and the study area contains many small, internally draining depression, a flood-fill model is a suitable approximation for extrapolation over short (~100m) distances.

Specific to this contribution, extrapolation serves an important role in cross-site comparison. In a low-relief environment we can expect discrete in flood connectivity, and categorical disagreements between sites reveal missing dynamics, namely storm-water infrastructure. The main point is that using flood fill models to propagate flood extents from multiple cameras improves our ability to identify these dynamics.

Given the widespread use of static water-level models for rapid flood assessment, it is valuable to discuss their behavior and limitations in the context of emerging flood data sources such as cameras (Gallien et al., 2014; Hong et al., 2024; Li et al., 2022; Preisser et al., 2022). We have expanded the discussion to highlight the limitations of the flood-fill approach and suggest avenues for integrating data from multiple cameras in future work.

> **Lines 787-808**: *"Flood fill methods are well-suited for short duration pluvial events in low relief, urban areas. Because the study sites within a self-contained depression, it is unlikely that there are substantial gradients in water surface elevation. This is supported by the 2D model results which, exclusive of edge effects, predicts a difference in water elevation between Sites A and B of only 0.5 cm after initial merging of the flood patches. Static elevation-based methods are widely used for rapid flood mapping, including in emergency management contexts (Gallien et al., 2024; Wang et al. 2024; Zheng et al., 2018). The cross-camera comparison used in this study is an effective tool for identifying potential failure modes within these models."*

3. Validation of the new method

The study would also be strengthened if estimated water surface elevations could be validated using other data sources. I understand that depth measurements may not be available, but could the authors estimate depth at strategic locations based on visible markers and compare those to estimates from their approach for the corresponding location? Also, some expanded discussion of uncertainty as a function of distance from the camera location would be beneficial.

Independent depth measurements are not available for the study site, and indeed the lack of such data is the primary motivation behind this project. However, there are identifiable markers of discrete jumps in water level, such as road overflow points (Vandele et al. 2019). Based on the lidar DTMs, and aerial imagery we compared projected flood extents, with road elevation profiles above and below spillover. This is included in a revised supplementary Figure:

[Figure]

"*SI 7: Road topographic profiles based on the 0.5m USGS DTM (black), 0.005m terrestrial DTM (red), and 0.5m terrestrial DTM (green). b) Flood extent interpolated from intersection of the projected terrestrial lidar points with the flood mask. C) Corresponding image with flood mask overlay.*

*To qualitatively validate the water level extraction method we examined observation immediately above and below overtopping of the road boundary. These were compared against cross-road elevation profiles extracted from both USGS and terrestrial DTMs. Prior to spillover, the projected flood extent ends at the road boundary, with water level slightly below the curb elevation found in the terrestrial DTM profile. After spillover, the project extent expands to fill the small paved area above the road surface, before stopping at the lawn boundary. This is consistent with the image observation, and topographic profile of a second spillover onto the lawn itself. Further from camera, toward the NW, decreased pixel resolution leads to the projected extent bleeding beyond the road, potentially upwardly biasing estimated water levels. The extracted water level is approximately 3 cm higher than the elevation contour best aligned with the projected flood boundary below spillover, and approximately 1cm higher after spillover.*

*The magnitude of both these biases decreases with larger flood extents due to more gradual elevation gradients, and the lack of curb shadows."*

We have added discussion of distance dependent pixel resolution and its influence on camera pose, and water level estimation:

> *Line 332-355: "Because pixel resolution decreases with distance from the camera, multiple 3D points may project onto a single image pixel; therefore, all inundated points are retained and a one-to-one pixel–point correspondence is not enforced. Together, these inundated points represent the portion of the ground surface that is underwater at the time the image was captured. This set of inundated points is interpolated into a 0.05-meter resolution raster representing the visible flood extent in the image. This interpolation step reduces bias associated with distance-dependent differences in point density and avoids over-representation of regions where many 3D points project to a single pixel. Water surface elevation (WSE) is estimated from the rasterized flood boundary rather than from individual pixels or raw point projections. Canny edge detection is applied to the rasterized inundation extent to identify the flood boundary, and the 90th and 95th percentiles of the resulting edge elevation distribution ($WSE_{90}$ and $WSE_{95}$) to account for potential topographic noise or obstruction of the water edge in the time lapse images. Assuming a flat water surface, elevations along the flood boundary should exhibit a sharp peak at the upper end of the elevation distribution. The consistency and sharpness of this peak are another parameter useful to evaluate the camera pose estimation, as errors in estimated camera orientation or translation produce unrealistically large elevation differences between near- and far-field water edges."*

Other comments

Figure 1b – this image is difficult to interpret, perhaps change the color scheme?

**[Q7]** We will adjust this figure to accentuate the small-scale variation of the floodplain, while preserving the context of the upland bluffs.

Section 2.3 – I recommend revising this section. It is difficult to follow the detailed accounts of start and end times. It might be better to display this as a figure or omit some of the detail not necessary for understanding the larger picture.

**[Q8]** We will add additional annotation of key elements of flood timing and duration to figures 3 & 4 and will remove unnecessary details from the text. However, we feel that some narrative description of the events as observed by the cameras is necessary to build reader intuition and understanding and better prime them for presentation of the flood extent estimates.

[Figure]

Figure 3: (a) SOFI time series for 14 May moderate severity case study event. Representative flooded images from (b) Site A and (c) Site B. Segmented flood masks are shown in blue.

[Figure]

Figure 4: (a) SOFI time series for 04 July severe case study event. Representative flooded images from (b) Site A and (c) Site B. Segmented flood masks are shown in blue.

Line 322: What do the authors mean by "rainfall was uniformly applied to the domain"? Precipitation data from one gauge was applied to the entire model domain? Uniform intensity? Please clarify.

**[Q9]** A single rain gauge record was used for each flood event, and that hyetograph was applied to the entire model domain grid, with no spatial variability in precipitation. While a simplification, the dominance of local pluvial runoff in the study area, and the short duration of the

case-study flood events, likely limits the influence of watershed scale precipitation gradients, and would not alter the basic behaviors of the rain-on-grid model relevant for comparison to the camera-derived estimates. We have clarified this in the methods:

> *Lines 350-353: "This model is implemented using the Hydrologic Engineering Center's River Analysis System (HEC-RAS), configured with a "rain-on-grid" unsteady boundary condition to simulate overland water flow across an 89.6 km² model domain covering the study site (USACE, 2022). The base terrain is the 0.5 m USGS DTM. Rainfall records defined the unsteady inputs the model domain, assuming spatially uniform precipitation."*

Line 365: Please explain how SOFI values should be interpreted.

**[Q10]** SOFI is the fraction of the total fraction of an image classified as flooded. It is included as a semi-quantitative metric of flood magnitude to contextualize the estimated water level, and flood extent trends. However, the absolute values of SOFI are a function of both the physical flood extent, and the perspective of the camera. For example, a camera installed directly Infront of a flood source will see SOFI initially rise very quickly, before leveling off as flooding fills the FOV (see SI Figure 2). We have added additional text explaining this interpretation:

> *Line 255-259"This ratio is referred to as the Static Observer Flooding Index (SOFI), following the approach of Vitry et al. (2019), providing a simple proxy for flood intensity as seen from a fixed observation point. SOFI has been shown to correlate strongly with changes in water level for a given location (Moy de Vitry et al. 2019). The shape and magnitude of SOFI response depend strongly on the geometry of a camera relative flooding, and as such values cannot be directly compared between study sites."*

Reply Citations:

Hong, Y., Kessler, J., Titze, D., Yang, Q., Shen, X., & Anderson, E. J. (2024). Towards efficient coastal flood modeling: A comparative assessment of bathtub, extended hydrodynamic, and total water level approaches. Ocean Dynamics, 74(5), 391-405.

Gallien, T. W., Sanders, B. F., & Flick, R. E. (2014). Urban coastal flood prediction: Integrating wave overtopping, flood defenses and drainage. Coastal Engineering, 91, 18-28.

Li, Z., Mount, J., & Demir, I. (2022). Accounting for uncertainty in real-time flood inundation mapping using HAND model: Iowa case study. Natural Hazards, 112(1), 977-1004.

Silverman, A. I., Brain, T., Branco, B., sai venkat Challagonda, P., Choi, P., Fischman, R., ... & Toledo-Crow, R. (2022). Making waves: Uses of real-time, hyperlocal flood sensor data for emergency management, resiliency planning, and flood impact mitigation. *Water Research*, *220*, 118648.

Gold, A., Anarde, K., Grimley, L., Neve, R., Srebnik, E. R., Thelen, T., ... & Hino, M. (2023). Data from the drain: A sensor framework that captures multiple drivers of chronic coastal floods. Water Resources Research, 59(4), e2022WR032392.

Negri, R., Ceferino, L., & Cremen, G. (2025). Prioritizing urban areas for the deployment of hyperlocal flood sensors using stakeholder elicitation and risk analysis. Natural Hazards Review, 26(3), 04025020.